# RegMean++: Enhancing Effectiveness and Generalization of Regression Mean for Model Merging

## Abstract

Model merging aims to combine task-specific models into a unified model that is capable of multi-tasking, without any computational overhead of re-training. Regression Mean (RegMean), an approach that formulates model merging as a linear regression problem, aims to find the optimal weights for each linear layer in the merge model by minimizing the discrepancy in predictions between the merge and candidate models. RegMean provides a precise closed-form solution for the merging problem; therefore, it offers explainability and computational efficiency. However, RegMean merges each linear layer independently, overlooking how the features and information in the earlier layers propagate through the layers and influence the final prediction in the merge model. In this paper, we introduce *RegMean++*, a simple yet effective alternative to RegMean, that explicitly incorporates both *intra- and cross-layer dependencies between merge models' layers* into RegMean's objective. By accounting for these dependencies, RegMean++ better captures the behaviors of the merge model. Extensive experiments demonstrate that RegMean++ consistently outperforms RegMean across diverse settings, including in-domain (ID) and out-of-domain (OOD) generalization, sequential merging, large-scale tasks, and robustness under several types of distribution shifts. Furthermore, RegMean++ achieves competitive or state-of-the-art performance compared to various recent advanced model merging methods.

## 1 Introduction

As pretrain-finetune paradigm becomes the foundation of modern machine learning, the number of pre-trained and fine-tuned task-specific models (candidate models) is growing at an unprecedented pace. Model merging (Matena & Raffel, 2022; Wortsman et al., 2022; Ilharco et al., 2022; Jin et al., 2022; Yadav et al., 2023; Yang et al., 2024b;a; Yadav et al., 2024), an emerged approach that aims to combine multiple candidate models into a single unified model (merge model) with multi-tasking capabilities, without incurring the computational overhead of traditional multi-task learning (MTL) or full access to original training data.

Regression Mean (RegMean; Jin et al. (2022)), an explainable and computationally efficient model merging method, formulates weight fusion as a closed-form regression problem that minimizes the differences between the outputs of merge model and those of each candidate model. RegMean leverages the inner-product matrices of the input features at each *linear layer*, including those within the *MLP components and attention heads* of transformer layers in the candidates. Additionally, RegMean decreases non-diagonal entries in these matrices to enhance stability during merging. The other types of transformer layers' weights are merged by simply averaging across candidate models. RegMean offers several benefits, including privacy-preserving, computational efficiency, and model-agnostic. These advantages make RegMean one of the most practical merging methods.

However, we discuss an important caveat of RegMean is that it operates by *independently* applying the closed-form solution to linear layers of the transformer layer, *fundamentally ignoring how features and information are processed and propagated through layers in the merge model, preventing it from generalizing well*. These intra- and cross-layer dependencies are crucial for maintaining good representations that influence the final predictions of the merge model.

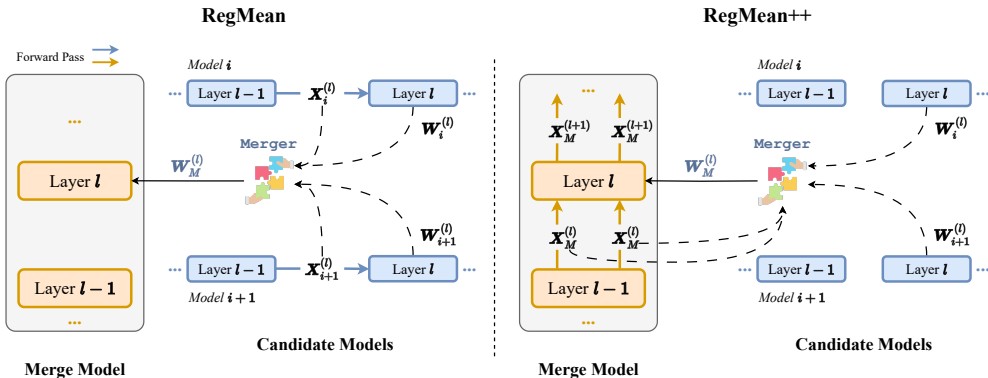

Figure 1: Comparison between RegMean and RegMean++ for merging. RegMean++ leverages representations from the merge model for merging, enabling accurate alignment with its behavior.

In this paper, we make the following contributions:

(1) We introduce *RegMean++*, a generalized extension of RegMean applicable to both vision and language tasks. RegMean++ incorporates both intra- and cross-layer dependencies of the merge model's layers into the RegMean merging objective. RegMean++ consistently improves in ID performance, OOD generalization, and demonstrates sustainability to sequential merging or large-scale tasks. Moreover, RegMean++ shows stronger robustness under various types of distribution shifts with reduced representation bias.

(2) We conduct layer-wise analysis and find that (i) merging using linear layers from middle and deep transformer layers preserves over $98\%$ accuracy compared to using all linear layers. In contrast, earlier transformer layers appear less important for merging, as using linear layers in them leads to significant accuracy degradation. (ii) Mid-depth layers consistently surpass the last layer in merging performance. This result highlights that the mid-depth layers may serve as more reliable sources of meaningful features for model merging. (iii) Merging using linear layers in the MLP modules consistently outperforms using linear layers in the attention heads.

(3) We benchmark RegMean++ against eleven advanced model merging methods. Experiment results show that RegMean++ achieves competitive or state-of-the-art performance across diverse settings, highlighting its generalization and effectiveness relative to existing approaches.

## 2 BACKGROUND AND RELATED WORK

**Model merging.** Model Soups (Wortsman et al., 2022; Choshen et al., 2022), a well-known approach that performs merging by simply taking the average of all candidate models' parameters, for enhancing distribution robustness (Wortsman et al., 2022), creating merge models with multi-task or multi-modality capabilities (Sung et al., 2023; Ilharco et al., 2022; Yadav et al., 2023).

Task Arithmetic (Ilharco et al., 2022) introduces *"task vector"*, a new concept that quantifies the task-specific knowledge of a candidate model, by measuring the difference between the candidate model parameters and the base model parameters. However, Yadav et al. (2023) point out that Task Arithmetic suffers from conflicts where different task vectors update the same parameters in opposite directions. They proposed TIES-Mering, a three-step method that first removes low-magnitude (noisy) parameters, then resolves sign conflicts, and finally aggregates only the non-conflicting parameters. DARE (Yu et al., 2023) proposes to pre-process each task vector independently by applying a Bernoulli mask to randomly zero out its parameters, effectively performing a dropout-like pruning before merging. TSV-M (Gargiulo et al., 2025) extracts task singular vectors via singular value decomposition (SVD) on layer-wise weight updates, selects top-$k$ components for a low-rank representation, and applies whitening to decorrelate subspaces before merging. Marczak et al. (2025) argue that effective merging depends on how well merge updates span each task's principal subspaces. They propose Iso-C, which sums task-specific updates, performs SVD, and replaces sin-

gular values with their average to create an isotropic spectrum. Iso-CTS, an Iso-C's variant, adds top singular directions from each task's residual, then orthogonalizes and isotropically scales the result.

Inspired by test-time adaptation schemes (Wang et al., 2020; Liang et al., 2025), AdaMerging (Yang et al., 2024b) adaptively learns the merging coefficient for each layer of each task vector by minimizing the entropy on unlabeled test data, using it as a surrogate objective to refine the merge model's performance across multiple tasks. DOGE AM (Wei et al., 2025) applies adaptive projective gradient descent at test time by jointly tuning a small modification vector and layer-wise merging coefficients to minimize prediction entropy on unlabeled inputs. DOGE AM projects updates orthogonally to a shared subspace to resolve task conflicts without hurting the shared knowledge.

Other approaches rely on data statistics, such as Fisher Merging (Matena & Raffel, 2022) that requires computing Fisher information matrices. RegMean (Jin et al., 2022) gets rid of expensive gradient computation, formulates merging as a regression problem, then comes up with a computationally efficient and explainable closed-form solution.

**Notations and problem formulation.** We denote matrices by boldface uppercase letters (*e.g.,* $\boldsymbol{X}$, $\boldsymbol{W}$, $\boldsymbol{\Lambda}$), scalars by lowercase letters (*e.g.,* $\alpha$, $l$). For operators, we denote $|| \cdot ||$ the Euclidean norm and $\mathrm{tr}(\cdot)$ the trace of a matrix. Following Jin et al. (2022), we consider the *training-free merging* framework. In such scenarios, we have access to a pool of multiple candidate models. Each candidate model is denoted as $f_i : \mathbb{R}^{N_i \times d} \to \mathbb{R}^{N_i \times |C|}$, and is fine-tuned on a task-specific dataset $\mathcal{D}_i = \{(\boldsymbol{X}_i, \boldsymbol{Y}_i)\}$. Here, $\boldsymbol{X}_i \in \mathbb{R}^{N_i \times d}$ is the batch-input of $N_i$ samples each of dimensionality $d$; and $\boldsymbol{Y}_i \in \mathbb{R}^{N_i \times |C|}$ is the corresponding target outputs, where $C$ is the set of classes. Our target is to find a merging function that takes a set of $K$ candidate models $f_i$, $i \in [1..K]$ and some data points, *e.g.,* the task-specific training samples or held-out out-of-domain samples, as the inputs and returns a merge model $f_M$. We assume that the model architecture for all models is the same.

**Regression Mean (RegMean;** (Jin et al., 2022)) formulates merging as an optimization problem that minimizes the prediction differences between the merge model and candidate models. More concretely, for each linear layer in a transformer layer $l$ of candidate model $f_i$, denoted as $\boldsymbol{W}_i^{(l)}$, given the input feature $\boldsymbol{X}_i^{(l)}$, RegMean minimizes the following regularized loss:

$$\mathcal{L}^{\text{RegMean}} = \sum_{i=1}^{K} \underbrace{||\boldsymbol{X}_i^{(l)} \boldsymbol{W}_M^{(l)} - \boldsymbol{X}_i^{(l)} \boldsymbol{W}_i^{(l)}||^2}_{①} + \sum_{i=1}^{K} \underbrace{\mathrm{tr}\left[(\boldsymbol{W}_M^{(l)} - \boldsymbol{W}_i^{(l)})^\top \boldsymbol{\Lambda}_i^{(l)}(\boldsymbol{W}_M^{(l)} - \boldsymbol{W}_i^{(l)})\right]}_{②}, \quad (1)$$

where $\boldsymbol{W}_M^{(l)}$ is the merge linear layer's weights at the same position as $\boldsymbol{W}_i^{(l)}$ in the model $f_i$, $\boldsymbol{\Lambda}_i^{(l)} = \frac{1-\alpha}{\alpha}\mathrm{diag}\left((\boldsymbol{X}_i^{(l)})^\top \boldsymbol{X}_i^{(l)}\right) \succeq 0$ is a regularization-strength diagonal matrix for $\boldsymbol{W}_i$, where $0 \leq \alpha \leq 1$ is a predetermined scaling factor. Minimizing the loss $\mathcal{L}^{\text{RegMean}}$ in Eqn. 1 means finding $\boldsymbol{W}_M^{(l)}$ that ① approximates the behaviors of all candidate models while ② enforcing a regularization that keeps linear layer's weights of the merge model $\boldsymbol{W}_M^{(l)}$ close to those of the candidate model $\boldsymbol{W}_i^{(l)}$. Minimizing $\mathcal{L}^{\text{RegMean}}$ describes a linear regression problem, where the inputs are $[\boldsymbol{X}_1^{(l)}, ..., \boldsymbol{X}_K^{(l)}]$ and the target outputs are $[\boldsymbol{X}_1^{(l)} \boldsymbol{W}_1^{(l)}, ..., \boldsymbol{X}_K^{(l)} \boldsymbol{W}_K^{(l)}]$. This objective has a closed-form solution as:

$$\boldsymbol{W}_M^{(l)} = \left[\sum_{i=1}^{K}\left(\widehat{\boldsymbol{G}}_i^{(l)}\right)\right]^{-1} \sum_{i=1}^{K}(\widehat{\boldsymbol{G}}_i^{(l)})\boldsymbol{W}_i^{(l)}, \quad (2)$$

where $\widehat{\boldsymbol{G}}_i^{(l)} = \alpha \boldsymbol{G}_i^{(l)} + (1-\alpha)\mathrm{diag}(\boldsymbol{G}_i^{(l)}) = \alpha(\boldsymbol{X}_i^{(l)})^\top \boldsymbol{X}_i^{(l)} + (1-\alpha)\mathrm{diag}((\boldsymbol{X}_i^{(l)})^\top \boldsymbol{X}_i^{(l)})$. Proof can be found in Appendix A. Other types of weights in the transformer layer are merged using averaging.

## 3 THE REGMEAN++

### 3.1 MOTIVATION

Our first start is to revisit the underlying RegMean's merging mechanism. RegMean operates by **independently** applying its closed-form solution to linear layers, including those within MLPs (up

and down projections) and attention heads (key, query, and value matrices), across all candidate models. These components are well-known to store most of the model's learned knowledge (Meng et al., 2022), which may explain the effectiveness of RegMean.

Deep networks consist of multiple non-linear components, such as GELU and LayerNorm, which are interleaved with linear components. Due to the non-linear properties, even a small change in the input might potentially cause a large, unpredictable shift in the output. RegMean overlooks the information flow and feature transformations occurring at both the **intra-layer level**, *i.e.,* within each layer, and **cross-layer level** of the merge model.

We hypothesize that incorporating those inference dynamics, that is, intra- and cross-layer dependencies, into RegMean's merging objective is crucial for improving the merge model's utilities and generalization. Inspired by this discussion, we introduce RegMean++, a simple yet powerful extension of RegMean in Section 3.2 below.

## 3.2 REGMEAN++ FOR MODEL MERGING

Let us fine-grain denote $X_i^{(l,j)}$ be the input features of the $j$-th linear layer $W_i^{(l,j)}$ in model $f_i$ at transformer layer $l$. RegMean computes the $j$-th merge linear layer weights $W_M^{(l,j)}$ by Eqn. 2. In this closed-form solution, the merge weight $W_M^{(l,j)}$ is determined by the individual weights and data statistics $G_i^{(l,j)} = (X_i^{(l,j)})^\top X_i^{(l,j)}$, which captures the dependencies among input features across all *candidate models*.

**Algorithm.** Similar to RegMean, given a $j$-th linear layer at the transformer layer $l$, RegMean++ computes the inner product matrix as $G_i^{(l,j)} = (X_i^{(l,j)})^\top X_i^{(l,j)}$. The key difference between RegMean++ and RegMean lies in how input feature $X_i^{(l,j)}$ is obtained: *For input features that are **activations** (cushion representations between transformer layers), RegMean++ computes $X_i^{(l,j)}$ based on the activations produced by the **previous merge layer** $f_M^{(l-1)}$ in the merge model, that is, $X_i^{(l)} = $*

---

**Algorithm 1** RegMean++ for Model Merging

**Require**: A pool of candidate models $f_i$, $i \in [1...K]$, a backbone for merging $f_M$, input embeddings, and a scaling factor $0 \leq \alpha \leq 1$.
**Ensure**: Return the merge model $f_M$.
1: **for** layer $l \in [1...L]$ **do**
2:     **for** linear layer $j \in [1...J]$ **do**
3:         Get input features of linear layer in $K$ candidate models: $X_1^{(l,j)}, X_2^{(l,j)}, ..., X_K^{(l,j)}$.
4:         **if** $X_i^{(l,j)}$ are activations from layer $l-1$ **then**
5:           $X_i^{(l,j)} \leftarrow f_M^{(l-1)}(X_i^{(l-1,j)})$
6:         **end if**
7:         Get linear layer weights in $K$ candidate models: $W_1^{(l,j)}, W_2^{(l,j)}, ..., W_K^{(l,j)}$.
8:         Compute $\widehat{G}_i^{(l,j)} \leftarrow \alpha G_i^{(l,j)} + (1 - \alpha)\text{diag}(G_i^{(l,j)})$
9:         Compute $W_M^{(l,j)}$ by Eqn. 2.
10:     **end for**
11:     Other weights are merged via averaging.
12: **end for**
13: **return** $f_M$

---

$f_M^{(l-1)}(X_i^{(l-1)})$ *while RegMean relies on the activations produced by the **previous candidate layer** $f_i^{(l-1)}$ in the candidate model, that is, $X_i^{(l)} = f_i^{(l-1)}(X_i^{(l-1)})$.* Similar to RegMean, all other parameters in the transformer layer, such as embeddings and biases, are merged via simple averaging. RegMean++ inherits RegMean's advantages, but introduces a trade-off between model performance and computational cost. During the statistic-collection phase, RegMean++ incurs additional forward passes, which are computed in the merge model to collect the inner-product matrices, yet the merging time equals that of RegMean. Our RegMean++ pseudocode is described in Algorithm 1. Comparison between RegMean and RegMean++ is described in Figure 1.

# 4 EXPERIMENTS

## 4.1 MODELS AND DATASETS

**Vision classification tasks.** Following prior works (Ilharco et al., 2022; Yang et al., 2024b; Wei et al., 2025), we evaluate the effectiveness of multi-task model merging methods on eight standard datasets including: SUN397 (Xiao et al., 2016), Stanford Cars (Cars) (Krause et al., 2013),

| Method | SUN397 | Cars | RESISC45 | EuroSAT | SVHN | GTSRB | MNIST | DTD | **Avg.** |
|---|---|---|---|---|---|---|---|---|---|
| Fine-tuned | 75.0 | 78.3 | 95.2 | 99.0 | 97.3 | 98.9 | 99.6 | 79.7 | 90.3 |
| MTL | 72.3 | 76.6 | 92.2 | 97.9 | 95.5 | 97.7 | 99.3 | 77.7 | 88.6 |
| *Data-Free Methods* | | | | | | | | | |
| Model Soups | 65.4 | 62.4 | 70.6 | 75.7 | 64.5 | 55.0 | 86.3 | 50.6 | 66.3 |
| Task Arithmetic | 57.0 | 55.7 | 64.7 | 73.3 | 77.9 | 68.5 | 96.1 | 47.1 | 67.5 |
| TIES-Merging | 67.0 | 64.2 | 74.3 | 74.5 | 77.7 | 69.4 | 94.1 | 54.0 | 71.9 |
| TSV-M | 67.6 | 71.6 | 84.7 | 93.4 | 91.9 | 92.5 | 98.9 | 63.8 | 83.1 |
| DOGE TA | 67.7 | 69.9 | 81.9 | 89.8 | 86.2 | 86.8 | 98.3 | 63.8 | 80.6 |
| Iso-C | 71.0 | 73.9 | 86.0 | 89.6 | 84.8 | 90.8 | 98.2 | 65.8 | 82.5 |
| Iso-CTS | **71.1** | **74.6** | 86.6 | 89.1 | 83.4 | 90.4 | 98.1 | 68.5 | 82.7 |
| *Training-Free Methods* | | | | | | | | | |
| Fisher Merging | 67.4 | 67.6 | 75.4 | 70.5 | 76.5 | 62.2 | 87.9 | 55.3 | 70.3 |
| RegMean | 68.6 | 70.0 | 84.6 | 95.4 | 92.6 | 83.4 | 98.4 | 66.1 | 82.4 |
| **RegMean++ (Ours)** | 69.3 | 70.5 | 86.7 | **96.1** | **94.1** | 90.4 | **99.0** | 68.7 | 84.4 |
| *Test-Time Adaption* | | | | | | | | | |
| Layer-wise AdaMerging | 67.8 | 71.1 | 83.9 | 92.3 | 87.8 | 93.3 | 98.2 | 66.8 | 82.6 |
| DOGE AM | 70.6 | 74.5 | **88.7** | 93.7 | 91.4 | **95.5** | 98.8 | **73.0** | **85.8** |

Table 1: Performance of all merging methods for ViT-B/32 measured on the 8-task benchmark. The **global best**, local best, and global runner-up are marked. See Appendix Table 11 and Table 12 for the results of ViT-B/16 and ViT-L/14.

RESISC45 (Cheng et al., 2017), EuroSAT (Helber et al., 2019), SVHN (Netzer et al., 2011), GT-SRB (Stallkamp et al., 2011), MNIST (LeCun et al., 1998), and DTD (Cimpoi et al., 2014). Following Wang et al. (2024), we also employ 12 additional tasks for evaluation on the sustainability.

We assess the performance of merging methods on three CLIP model variants (Radford et al., 2021) with ViT-B/32, ViT-B/16, and ViT-L/14. For the candidate models, we employ off-the-shelf checkpoints from Tang et al. (2024a).

**Language generation tasks.** Following He et al. (2025), we evaluate the merge models on 11 datasets reflecting five domains: (1) *instruction following:* IFEval (Zhou et al., 2023), (2) *mathematics:* GSM8K (Cobbe et al., 2021), (3) *multilingual understanding* (on French, Spanish, German, and Russian): Multilingual MMLU, Multilingual ARC, and Multilingual Hellaswag (Lai et al., 2023), (4) *coding:* HumanEval+ and MBPP+ (Liu et al., 2023), and (5) *safety:* WildGuardTest (Han et al., 2024), HarmBench (Mazeika et al., 2024), DoAnythingNow (Shen et al., 2024), and XSTest (Röttger et al., 2024).

We assess the performance of merging methods on two Llama 3 variants (Grattafiori et al., 2024) with Llama-3.2-3B and Llama-3.1-8B. For the candidate models, we employ off-the-shelf checkpoints from He et al. (2025).

Details of these datasets' description and models can be found in Appendix B.1 and Appendix B.2.

## 4.2 COMPARISON METHODS

We compare RegMean++ against 11 recent advanced model merging methods spanning three categories: (1) *Data-Free methods:* Model Soups (Wortsman et al., 2022), Task Arithmetic (Ilharco et al., 2022), TIES-Merging (Yadav et al., 2023), TSV-M (Gargiulo et al., 2025), DOGE TA (Wei et al., 2025), Iso-C and Iso-CTS (Marczak et al., 2025). (2) *Training-Free methods:* Fisher Merging (Matena & Raffel, 2022) and RegMean (Jin et al., 2022). (3) *Test-Time Adaptation:* AdaMerging (Yang et al., 2024b) and DOGE AM (Wei et al., 2025). In addition, we also consider the MTL as an upper-bound performance. Details can be found in Appendix B.3.

## 4.3 EXPERIMENTAL SETUP

We employ FusionBench (Tang et al., 2024a) and MergeBench (He et al., 2025) for merging evaluation on vision and language tasks, respectively. For vision tasks, we report the *accuracy* and *normalized accuracy* (See Appendix C.1 for details). For language tasks, multiple *task-specific metrics* are employed (See Appendix C.2 for details). All of the metrics are calculated on the tasks' test

split. Hyperparameters for settings are specified in their respective subsections, and further detailed in Appendix C.3. Due to space constraints, we present key results in the main text and defer full experimental setups and additional results to Appendix C.4 and Appendix D, respectively. Implementation and guidelines for reproducing those results are attached to the supplemental materials.

## 5 RESULTS AND ANALYSIS

### 5.1 MAIN RESULTS

Performance of RegMean++ and other methods on eight vision tasks is shown in Table 1. We observe that RegMean++ consistently surpasses RegMean on all tasks, achieving an average improvement of 2.0%, and demonstrates gains of 1.2% and 0.6% when evaluated on ViT-B/16 and ViT-L/14, respectively. Compared to other data-free and training-free methods, RegMean++ achieves competitive or the best performance. Furthermore, RegMean++, despite requiring no access to test-time data or optimization, can rival or surpass test-time adaptation methods: outperforming Layer-wise AdaMerging (84.4% vs. 82.6%), and close to DOGE AM (85.8%). RegMean++ achieves the best results on three specific tasks—EuroSAT (96.1%), SVHN (94.1%), and MNIST (99.0%). These results validate the significant advantage of leveraging the intra- and cross-layer dependencies.

Moreover, the performance gains come with reduced representation bias across non-linear components, consistent with observations from Yang et al. (2024a). As shown in Figure 2 for ViT-B/32, RegMean++ improves upon RegMean with stronger feature alignment between the merge and candidate models across tasks. This validates the importance of modeling feature flow in deep networks.

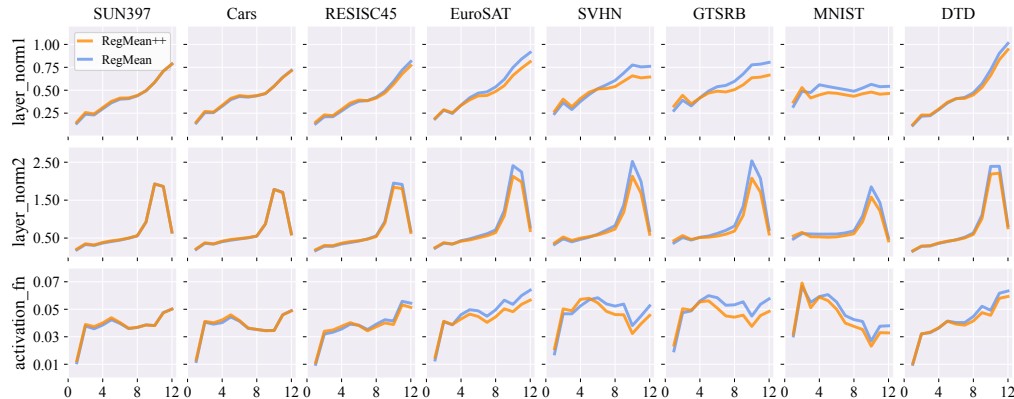

Figure 2: Representation bias in three non-linear components, namely two LayerNorms and a GELU activation, across transformer layers for RegMean and RegMean++ on 8-task merging with ViT-B/32 model. The representation bias is quantified on the task-specific test datasets. Corresponding visualizations for ViT-B/16 and ViT-L/14 are shown in Appendix Figure 9 and Appendix Figure 10.

### 5.2 SUSTAINABILITY TO LARGE-SCALE TASKS

In this section, following Wang et al. (2024), we evaluate the sustainability of merging methods when scaling the number of tasks up to 20. A higher number of merging tasks, a higher level of complexity and conflict between candidate models. Merging methods are thus expected to maintain high accuracy under this large-scale task setting. Since evaluating all possible task combinations is computationally expensive, we fix the order of tasks and add four tasks one by one. See Appendix C.4 for the experiment setting and details on the order of tasks. Note that these experiments are conducted independently; that is, for each algorithm, the merging process is re-executed from scratch whenever new tasks are added, and the performance is calculated only on the involved tasks. Performance of merging methods on vision tasks for ViT-B/32, ViT-B/16, and ViT-L/14 is visualized in Figure 3. We observe that RegMean++, along with RegMean, TSV-M, Iso-C, Iso-CTS, AdaMering, and DOGE AM demonstrate strong sustainability as the number of tasks increases. In contrast, Model Soups, Task Arithmetic, TIES-Merging, Fisher Merging, and DOGE TA exhibit a noticeable decline in accuracy as the number of tasks increases.

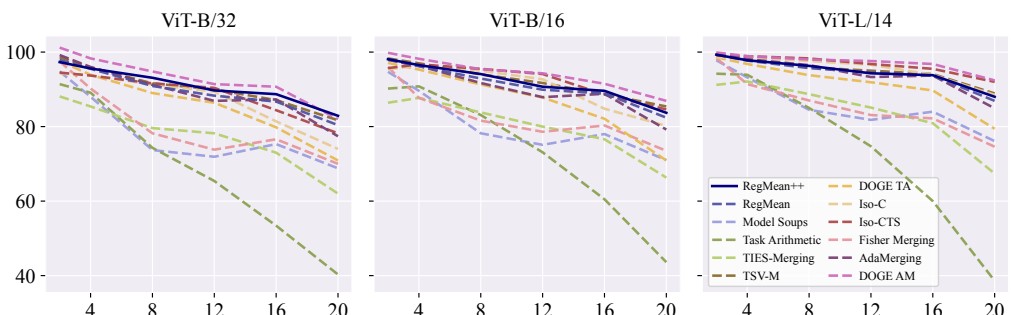

Figure 3: Average normalized accuracy of all merging methods for ViT-B/32, ViT-B/16, and ViT-L/14; evaluated on different numbers of tasks (up to 20 tasks).

## 5.3 SEQUENTIAL MERGING

Merging from scratch when new candidate models come is computationally expensive and infeasible. Different from one-time large-scale merging described in Section 5.2, sequential merging is a practical scenario where tasks arrive over time. In this setting, we evaluate the performance of RegMean and RegMean++ by merging the first four candidates in a predefined task sequence, then progressively merging the result model with the next four, repeating this process until all 20 tasks are merged.

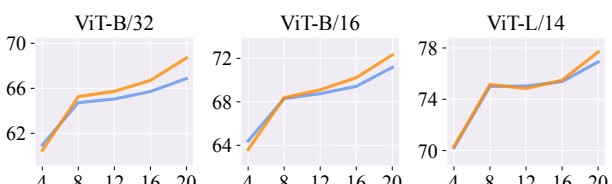

Figure 4: Sequential merging performance of RegMean++ (dark orange) and RegMean (cornflower blue) for ViT-B/32, ViT-B/16, and ViT-L/14. Results show the mean of average accuracy on all 20 tasks across five different task sequences.

Figure 4 presents the performance of RegMean and RegMean++ for all three vision models, averaged over five different task sequences. See Appendix C.4 for more details on the experiment setting. RegMean++ demonstrates greater improvements when more candidate models are merged. Especially for ViT-B/32 and ViT-B/16, where the performance gaps become more apparent as the number of merged tasks increases. Further analysis indicates that RegMean++ better accommodates new tasks while exhibiting reduced forgetting on the early ones, indicated in Appendix Figure 11.

We further present a comprehensive comparison of RegMean++ with other merging methods for ViT-B-32 model in Figure 5. We find that RegMean++ achieves superior performance after 8, 12, 16, and 20 tasks have been merged. Iso-C and Iso-CTS, although demonstrating strong performance on standard scenarios, fail dramatically in sequential merging. Their performance shows a sharp decreasing trend after 12 tasks are merged.

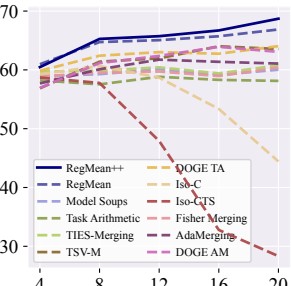

Figure 5: Sequential merging performance of all methods for ViT-B-32 model.

## 5.4 ROBUSTNESS AGAINST DISTRIBUTION SHIFTS

We first note that the notion of distribution shift is very broad and can exhibit in many forms, such as covariance shift, label shift, concept shift, class-conditional shift, etc. Here, following Tang et al. (2024b); Yang et al. (2024b); Hendrycks & Dietterich (2019), we evaluate merging methods' robustness on vision test data by employing seven types of noises (covariance shift), including Motion Blur, Impulse Noise, Gaussian Noise, Pixelate, Spatter, Contrast, and JPEG Compression. These noises are introduced into four datasets: Cars, EuroSAT, RESISC45, and GTSRB. Table 2 shows the average accuracy of merging methods for ViT-B/32 under corrupted test data. We observe that

RegMean++ achieves superior performance as it surpasses all training-free and data-free methods on both the clean and the corrupted test sets. A similar trend can also be observed for ViT-B/16 and ViT-L/14 in Appendix Table 13 and Table 14, respectively. These results highlight that RegMean++ not only performs effectively on ID and OOD tasks but is also robust to distribution shift.

| Method | Clean Test Set | Corrupted Test Set | | | | | | | |
|---|---|---|---|---|---|---|---|---|---|
| | | Motion | Impulse | Gaussian | Pixelate | Spatter | Contrast | JPEG | Avg. |
| *Data-Free Methods* | | | | | | | | | |
| Model Soups | 76.0 | 64.6 | 56.9 | 58.1 | 28.5 | 61.3 | 64.7 | 66.3 | 57.2 |
| Task Arithmetic | 77.5 | 65.9 | 58.9 | 59.6 | 29.7 | 63.5 | 66.0 | 67.8 | 58.8 |
| TIES-Merging | 73.3 | 63.2 | 54.5 | 56.2 | 28.1 | 57.7 | 63.8 | 64.4 | 55.4 |
| TSV-M | 88.3 | 78.9 | 69.9 | 69.1 | 37.8 | 75.4 | 77.2 | 80.2 | 69.8 |
| DOGE TA | 86.1 | 77.3 | 66.0 | 66.2 | 37.5 | 71.5 | 76.1 | 77.7 | 67.5 |
| Iso-C | 84.8 | 75.9 | 62.2 | 63.9 | 35.0 | 69.2 | 76.5 | 75.3 | 65.4 |
| Iso-CTS | 84.9 | 75.7 | 61.9 | 63.5 | 34.2 | 69.2 | 76.5 | 75.4 | 65.2 |
| *Training-Free Methods* | | | | | | | | | |
| Fisher Merging | 79.1 | 67.0 | 59.8 | 60.7 | 29.3 | 64.9 | 67.9 | 69.0 | 59.8 |
| RegMean | 89.1 | 79.7 | 69.1 | 67.3 | 37.1 | 75.2 | 78.0 | 80.9 | 69.6 |
| **RegMean++** (Ours) | 89.7 | 81.8 | 70.0 | 68.4 | 37.9 | **75.9** | 79.6 | 82.8 | 70.9 |
| *Test-Time Adaptation* | | | | | | | | | |
| Layer-wise AdaMerging | 88.4 | 81.3 | 69.0 | 71.2 | 41.3 | 74.5 | 80.2 | 80.1 | 71.1 |
| DOGE AM | **90.9** | **85.0** | 65.7 | **73.4** | **44.2** | 75.4 | **83.8** | **84.2** | **73.1** |

Table 2: Performance of merging methods of ViT-B/32 on corrupted test data. The **global best**, local best, and global runner-up are marked.

## 5.5 OUT-OF-DOMAIN GENERALIZATION

In this section, we evaluate the OOD generalization of merging methods. We randomly select two from eight vision tasks to serve as OOD tasks, while the remaining six are used for merging. Table 3 reports the ID and OOD accuracy of all merging methods evaluated on three ViT models. Each result is the mean of the average accuracy over five runs. Across all models, RegMean++ consistently achieves strong ID performance, outperforming RegMean by a margin of +1.6, +0.9, and +0.4 points for ViT-B/32, ViT-B/16, and ViT-L/14, respectively. RegMean++ slightly outperforms RegMean for all three models on OOD tasks. Notably, TIES-Merging achieves the best OOD accuracy across ViT-B/32 (51.7%), ViT-B/16 (59.9%), and ViT-L/14 (68.3%), but low performance

| Method | ViT-B/32 | | ViT-B/16 | | ViT-L/14 | |
|---|---|---|---|---|---|---|
| | ID | OOD | ID | OOD | ID | OOD |
| *Data-Free Methods* | | | | | | |
| Model Soups | 70.2 | 50.6 | 75.4 | 58.9 | 82.7 | 67.9 |
| Task Arithmetic | 73.7 | 42.6 | 80.2 | 54.1 | 84.7 | 61.8 |
| TIES-Merging | 73.0 | 51.7 | 77.7 | 59.9 | 84.7 | 68.3 |
| TSV-M | 84.8 | 45.7 | 88.2 | 54.8 | 91.3 | 62.1 |
| DOGE TA | 83.1 | 49.4 | 86.5 | 56.0 | 89.8 | 65.8 |
| Iso-C | 83.1 | 49.3 | 87.8 | 58.2 | 92.4 | 65.9 |
| Iso-CTS | 82.8 | 48.6 | 88.1 | 58.2 | 92.8 | 65.2 |
| *Training-Free Methods* | | | | | | |
| Fisher Merging | 74.3 | 51.0 | 78.4 | 58.0 | 83.7 | 64.2 |
| RegMean | 84.6 | 48.3 | 88.0 | 56.5 | 91.5 | 64.5 |
| **RegMean++** (Ours) | 86.2 | 48.9 | 88.9 | 56.7 | 91.9 | 64.7 |
| *Test-Time Adaptation* | | | | | | |
| Layer-wise AdaMerging | 84.5 | 48.9 | 87.2 | 55.5 | 91.8 | 65.3 |
| DOGE AM | **87.4** | 47.7 | **89.6** | 54.9 | **93.0** | 63.4 |

Table 3: ID and OOD performance of ViT-B/32, ViT-B/16, and ViT-L/14. We report the mean of the average accuracy over five runs. The **global best**, local best, and global runner-up are marked.

in ID tasks, highlighting *a trade-off between OOD and ID generalization in merging*. Overall, RegMean++ offers a good trade-off between ID and OOD generalization across all models, slightly outperforming RegMean, and demonstrating competitive performance without requiring computation or access to test data, as for test-time adaptation methods.

## 5.6 EFFECTS OF MERGING IN DIFFERENT SPACES

One might ask: (1) Which component—attention heads or MLPs—contributes more effectively to merging performance? (2) How does transformer-layer choice influence the merging effectiveness?

In this section, we measure the effects of merging in different spaces (*i.e.,* different layers and components) for RegMean++ and RegMean. We perform the following empirical experiments: (1) *region-specific merging:* all the linear layers from a specific set of transformer layers grouped by position in the model are used for merging and compare performance across three configs: (i) early layers $(1, 2, 3, 4)$, (ii) middle layers $(5, 6, 7, 8)$, and deep layers $(9, 10, 11, 12)$. (2) *Layer-wise merging:* all the linear layers from a specific transformer layer are used for merging. (3) *Component-specific merging:* all the linear layers in MLP components or attention heads are used for merging. Note that across these empirical experiments, only the selected linear layers are merged using the respective merging methods, while simple averaging is applied for the other linear layers.

We report the results on vision tasks in Table 4 and Figure 6. We defer additional results and analysis of other merging methods to Appendix D. Overall, in all merging scenarios—region-specific, layer-wise, and component-specific—RegMean++ consistently improves over RegMean. These results validate the enhanced capability of RegMean++ in exploiting the layer's dependencies in the merge model for better merging. Furthermore, we observe the following important insights.

**Merging using middle and deep layers preserves overall performance.** For all models and merging methods, using the middle and deep layers (layers 5-12 for ViT-B/32 and ViT-B/16, and layers 8-24 for ViT-L/14) achieves high performance, closely matching that of using all layers. For example, Reg-Mean++ achieves 98%, 99%, and 99% of the full-layer accuracy for ViT-B/32 $(83.5/84.4)$ and ViT-B/16 $(86.5/87.2)$, and ViT-L/14 $(90.5/91.0)$. When

| Components | ViT-B/32 | | ViT-B/16 | | ViT-L/14 | |
|---|---|---|---|---|---|---|
| | RegMean | RegMean++ | RegMean | RegMean++ | RegMean | RegMean++ |
| All | 82.4 | 84.4 | 86.0 | 87.2 | 90.4 | 91.0 |
| Early | 67.1 | 67.8 | 73.9 | 74.8 | 81.2 | 81.5 |
| Middle | 74.0 | **77.2** | **78.9** | **81.0** | 85.6 | **86.6** |
| Deep | **74.4** | 75.7 | 78.7 | 79.5 | **85.8** | 86.4 |
| Middle & Deep | 80.7 | 83.5 | 84.4 | 86.5 | 89.4 | 90.5 |
| Attention heads | 72.9 | 76.0 | 78.0 | 80.1 | 85.0 | 86.5 |
| MLPs | **78.8** | **81.4** | **82.7** | **84.5** | **88.1** | **88.9** |

Table 4: Region-specific merging and component-specific merging performance of RegMean++ and RegMean for ViT-B/32, ViT-B/16, and ViT-L/14 across eight tasks.

merging is performed using only the middle or deep layers, performance remains above 90% of the full-layer. In contrast, early layers contribute less, with a notable drop in accuracy.

**MLP module linear layer merging outperforms attention head merging.** Component-specific merging analysis shows that merging using linear layers from the MLP modules yields higher accuracy than that from attention heads across all models. This implies that MLPs may contain richer semantic representations for merging. This observation aligns with previous findings (Geva et al., 2021; Meng et al., 2022; Chen et al., 2024), which have shown that MLPs serve as dictionaries of factual and task-relevant knowledge in transformer models.

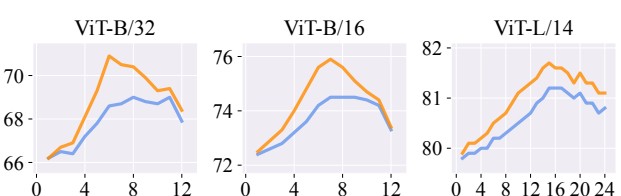

Figure 6: Layer-wise merging performance of Reg-Mean++ (dark orange) and RegMean (cornflower blue) for ViT-B/32, ViT-B/16, and ViT-L/14. Results show average accuracy across eight tasks.

**Intermediate (middle and deep) layers surpass the last layer in merging performance.** Figure 6 shows that middle layers consistently surpass the deep layers and the last layer for merging. This suggests that the deep layers and final layer may be specialized for task-specific, while the middle layers serve as sources of more meaningful features beneficial for merging.

## 5.7 EFFECTS OF DATA CHARACTERISTICS

Data plays the central role in the training-free merging framework. However, full access to training datasets is often restricted in practice due to privacy concerns. In this section, we investigate the ef-

fects of data characteristics on merging performance for vision tasks with three different experiment settings: (1) *number of samples:* we randomly select samples from the training set of each task, (2) *class imbalance:* we randomly select samples from a random class in the training set of each task, and (3) *OOD samples:* we randomly select samples from the ImageNet database (Deng et al., 2009) as an OOD dataset for all tasks.

**Effects of OOD samples.**  Table 5 indicates that if OOD samples are used for merging, Fisher Merging demonstrates a stable performance. In contrast, RegMean and RegMean++ exhibit reduced accuracy, particularly for ViT-B/32 (*e.g.,* RegMean++ drops from 84.4% to 65.5%). This reflects a limitation of regression-based methods when the merging data distribution is misaligned with the task domains.

| Characteristics | ViT-B/32 | | | ViT-B/16 | | | ViT-L/14 | | |
|---|---|---|---|---|---|---|---|---|---|
| | Fisher | RM | RM++ | Fisher | RM | RM++ | Fisher | RM | RM++ |
| ID (random) | 70.3 | 82.4 | 84.4 | 75.6 | 86.0 | 87.2 | 82.4 | 90.4 | 91.0 |
| Class Imbalance | 69.2 | 75.5 | 76.5 | 74.5 | 80.6 | 81.6 | 77.3 | 86.0 | 86.4 |
| OOD Samples | 68.5 | 66.5 | 65.5 | 73.9 | 70.2 | 71.9 | 80.6 | 78.9 | 79.6 |

Table 5: Accuracy of Fisher Merging (Fisher), RegMean (RM), and RegMean++ (RM++) for ViT-B/32, ViT-B/16, and ViT-L/14 on three scenarios: random ID samples, ID class imbalance, and OOD samples. We report average accuracy across eight tasks.

**Effects of the number of samples.**  Figure 7 and Appendix Figure 8 show that merging performance improves modestly as the ID sample count increases. The accuracy quickly saturates. This result implies that the effectiveness of merging is influenced more by the quality of the selected samples than by their quantity. Even using a small number of ID samples can achieve near-optimal performance in training-free settings.

**Effects of class imbalance.**  As shown in Table 5, RegMean++ performs the best in a such circumstance. However, Fisher Merging, by leveraging the class labels, remains relatively stable with the least accuracy drop compared to merging in an ideal scenario, *i.e.,* class balance.

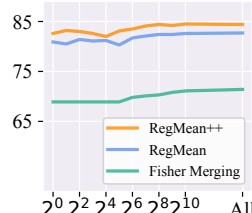

Figure 7: Impact of ID sample count on training-free merging methods for ViT-B/32.

## 5.8 Performance of RegMean++ on Language Tasks

Performance of RegMean and RegMean++ on language tasks is shown in Table 6, with each entry representing the average results across tasks in that domain. We perform grid search over scaling factors $\alpha \in \{0.1, 0.3, 0.5, 0.7, 0.9\}$, and report the best results based on the averaged multi-domain performance. To compute the inner-product matrices, we use 256 samples per domain with a max sequence length of 2048, uniformly drawn from the candidate models' original training sets provided by He et al. (2025). We find that RegMean++ outperforms RegMean when merging Llama-3.1-8B candidates, while underperforming when merging Llama-3.2-3B candidates. Further, RegMean++ underperforms RegMean in the instruction following task.

| Method | Instruction following | Math | Multilingual | Coding | Safety | Avg. |
|---|---|---|---|---|---|---|
| | | | *Llama-3.2-3B* | | | |
| RegMean | **8.3** | **35.5** | 47.3 | **39.2** | **39.8** | **34.0** |
| **RegMean++** | 6.8 | 35.1 | **47.4** | 37.0 | 38.5 | 33.0 |
| | | | *Llama-3.1-8B* | | | |
| RegMean | **26.6** | 63.2 | 49.0 | 48.3 | 36.9 | 44.8 |
| **RegMean++** | 11.1 | **65.8** | **53.1** | **52.3** | **46.3** | **45.7** |

Table 6: Comparison of RegMean and RegMean++ on language tasks with two Llama 3 variants.

## 6 Conclusion

This paper introduces RegMean++, a generalized extension of RegMean applicable to both vision and language tasks. RegMean++ incorporates both intra- and cross-layer dependencies of the merge model's layers into the RegMean merging objective. RegMean++ addresses RegMean's limitations in information flow and feature transformations. Extensive experiments demonstrate that RegMean++ achieves robust ID performance, improved OOD generalization, and strong scalability, outperforming or matching state-of-the-art merging methods across a wide range of settings.

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

# A DERIVATION OF REGMEAN'S REGULARIZED LOSS

Without loss of generality, we omit the notation transformer layer $l$ of the linear layer's weights and input features. For each linear layer of candidate model $f_i$, denoted as $\boldsymbol{W}_i$, given the input feature $\boldsymbol{X}_i$, RegMean minimizes the following regularized loss:

$$\mathcal{L}^{\text{RegMean}} = \sum_{i=1}^{K} ||\boldsymbol{X}_i \boldsymbol{W}_M - \boldsymbol{X}_i \boldsymbol{W}_i||^2 + \sum_{i=1}^{K} \text{tr}\left[(\boldsymbol{W}_M - \boldsymbol{W}_i)^\top \boldsymbol{\Lambda}_i (\boldsymbol{W}_M - \boldsymbol{W}_i)\right],$$

where $\boldsymbol{W}_M$ is the merge linear layer's weights at the same position as $\boldsymbol{W}_i$ in the model $f_i$, $\boldsymbol{\Lambda}_i = \frac{1-\alpha}{\alpha}\text{diag}\left(\boldsymbol{X}_i^\top \boldsymbol{X}_i\right) \succeq 0$ is a regularization-strength diagonal matrix for $\boldsymbol{W}_i$, where $0 \leq \alpha \leq 1$ is a predetermined scaling factor. This objective has a closed-form solution as:

$$\boldsymbol{W}_M = \left(\sum_{i=1}^{K} \widehat{\boldsymbol{G}}_i\right)^{-1} \sum_{i=1}^{K} \widehat{\boldsymbol{G}}_i \boldsymbol{W}_i,$$

where $\widehat{\boldsymbol{G}}_i = \alpha \boldsymbol{G}_i + (1-\alpha)\text{diag}(\boldsymbol{G}_i) = \alpha \boldsymbol{X}_i^\top \boldsymbol{X}_i + (1-\alpha)\text{diag}(\boldsymbol{X}_i^\top \boldsymbol{X}_i)$.

*Proof.* Take derivative of $\mathcal{L}^{\text{RegMean}}$ with respect to $\boldsymbol{W}_M$:

$$\begin{aligned}
\nabla_{\boldsymbol{W}_M} \mathcal{L}^{\text{RegMean}} &= \sum_{i=1}^{K} 2\boldsymbol{X}_i^\top (\boldsymbol{X}_i \boldsymbol{W}_M - \boldsymbol{X}_i \boldsymbol{W}_i) + \sum_{i=1}^{K} (\boldsymbol{\Lambda}_i(\boldsymbol{W}_M - \boldsymbol{W}_i) + \boldsymbol{\Lambda}_i^\top(\boldsymbol{W}_M - \boldsymbol{W}_i)) \\
&= \sum_{i=1}^{K} 2(\boldsymbol{X}_i^\top \boldsymbol{X}_i \boldsymbol{W}_M - \boldsymbol{X}_i^\top \boldsymbol{X}_i \boldsymbol{W}_i) + \sum_{i=1}^{K} 2\boldsymbol{\Lambda}_i(\boldsymbol{W}_M - \boldsymbol{W}_i) \\
&= \sum_{i=1}^{K} 2(\boldsymbol{X}_i^\top \boldsymbol{X}_i + \boldsymbol{\Lambda}_i)\boldsymbol{W}_M - \sum_{i=1}^{K} 2(\boldsymbol{X}_i^\top \boldsymbol{X}_i + \boldsymbol{\Lambda}_i)\boldsymbol{W}_i.
\end{aligned}$$

We see that $\mathcal{L}^{\text{RegMean}}$ is convex. Letting $\nabla_{\boldsymbol{W}_M} \mathcal{L}^{\text{RegMean}} = 0$, we can find the optimal $\boldsymbol{W}_M^*$ such that $\mathcal{L}^{\text{RegMean}}$ reaches the global minimum:

$$\boldsymbol{W}_M^* = \left[\sum_{i=1}^{K} (\boldsymbol{X}_i^\top \boldsymbol{X}_i + \boldsymbol{\Lambda}_i)\right]^{-1} \sum_{i=1}^{K} (\boldsymbol{X}_i^\top \boldsymbol{X}_i + \boldsymbol{\Lambda}_i)\boldsymbol{W}_i. \tag{3}$$

Substitute $\boldsymbol{\Lambda}_i = \frac{1-\alpha}{\alpha}\text{diag}\left(\boldsymbol{X}_i^\top \boldsymbol{X}_i\right)$ into Eqn. 3, we have:

$$\begin{aligned}
\boldsymbol{W}_M^* &= \left[\sum_{i=1}^{K} (\boldsymbol{X}_i^\top \boldsymbol{X}_i + \frac{1-\alpha}{\alpha}\text{diag}(\boldsymbol{X}_i^\top \boldsymbol{X}_i))\right]^{-1} \sum_{i=1}^{K} (\boldsymbol{X}_i^\top \boldsymbol{X}_i + \frac{1-\alpha}{\alpha}\text{diag}(\boldsymbol{X}_i^\top \boldsymbol{X}_i))\boldsymbol{W}_i \\
&= \left(\sum_{i=1}^{K} \widehat{\boldsymbol{G}}_i\right)^{-1} \sum_{i=1}^{K} \widehat{\boldsymbol{G}}_i \boldsymbol{W}_i,
\end{aligned}$$

which completes the proof. $\square$

# B DATASETS, MODELS, AND MERGING METHODS

## B.1 DATASETS

**Standard vision classification datasets.** Task descriptions and statistics of datasets used for the standard 8-task image classification benchmark are described below. These datasets are publicly available `https://huggingface.co/collections/tanganke/the-eight-image-classification-tasks`.

- **SUN397** (Xiao et al., 2016) contains more than $100,000$ images of 397 categories for benchmarking scene understanding. The number of images varies across categories, but there are at least 100 images each.

- **Stanford Cars (Cars)** (Krause et al., 2013) has $16,185$ images in total of 196 types of cars and evenly split for training and testing sets.

- **RESISC45** (Cheng et al., 2017) is developed for remote sensing image scene classification. This dataset covers 45 scene classes with 700 images of size $256 \times 256$ for each.

- **EuroSAT** (Helber et al., 2019) is used for land use and land cover classification using Sentinel-2 satellite images of size $64 \times 64$, consisting of $27,000$ images covering 10 classes.

- **SVHN** (Netzer et al., 2011) is a street view house number classification benchmark, containing more than $600,000$ RGB images of 10 printed digits in size $32 \times 32$ cropped from house number plates.

- **GTSRB** (Stallkamp et al., 2011) is a German traffic sign recognition benchmark consisting of over $50,000$ images of 43 classes of traffic signs in varying light and background conditions.

- **MNIST** (LeCun et al., 1998), a well-known classical dataset for hand-written digit classification with $60,000$ training and $10,000$ testing images of size $28 \times 28$ in 10 classes of numbers.

- **DTD** (Cimpoi et al., 2014) is a collection of $5,640$ images across 47 categories of textures in the wild, annotated with human-centric attributes.

**Additional vision classification datasets.** Besides the standard 8-task scenario, we follow the previous work and further extend our experimental scenario to 20 tasks. The new 12 tasks are listed below. These datasets are publicly available at `https://huggingface.co/collections/tanganke/image-classification-datasets`.

- **Flowers102** (Nilsback & Zisserman, 2008) contains 102 flower categories that are popular in the United Kingdom, with $1,020$ training and $6,149$ testing images. The images have varying poses and light conditions.

- **PCAM** (PatchCamelyon) (Veeling et al., 2018) consists of more than 300M color images in size of $96 \times 96$ pixels extracted from histopathologic scans of lymph node sections. Each of them is annotated with a binary class indicating the presence of metastatic tissue.

- **FER2013** (Goodfellow et al., 2013) is developed for facial expression recognition. The images are grayscale and have a size of $48 \times 48$ pixels, describing seven different kinds of emotions. The training and testing split consists of $28,709$ and $7,178$ samples, respectively.

- **OxfordIIITPet** (Parkhi et al., 2012) is a 37-category pet dataset with roughly 200 images for each category, and is equally divided for both training and testing splits. The images vary in scale, pose, and lighting conditions.

- **STL10** (Coates et al., 2011) is primarily built for unsupervised image recognition tasks covering 10 classes. Hence, the number of labeled images is quite small: 500 training and 800 testing images for each class. All of them are in $96 \times 96$ pixel resolution.

- **CIFAR100** (Krizhevsky et al., 2009) consists of color images categorized in 100 general classes, each class contains 600 images, and each image is in size $32 \times 32$. There are $50,000$ training images and $10,000$ testing images.

- **CIFAR10** (Krizhevsky et al., 2009) is similar to CIFAR100, except it has 10 classes.

- **Food101** (Bossard et al., 2014) contains of 101 food categories, with $101,000$ images. For each class, 750 images are for training and 250 are for testing. Only the testing images are manually reviewed. The training images contain noise mostly from intense colors, and sometimes are mislabelled.

- **FashionMNIST** (Xiao et al., 2017) is designed as a drop-in replacement benchmark for the original MNIST, thereby inheriting the same structure as MNIST.

- **EMNIST** (Cohen et al., 2017) is an extended version of MNIST. EMNIST contains images of both characters and digits. We choose to use only the EMNIST Letters split, which contains around $145,000$ images evenly distributed in 26 classes of the alphabet letters.

- **KMNIST** (Clanuwat et al., 2018), yet another version of MNIST, represents 10 Japanese Hiragana characters.

- **RenderedSST2** (Socher et al., 2013b; Radford et al., 2019) is used for evaluating the models' capability on optical character recognition. The images are rendered from sentences in the Stanford Sentiment Treebank v2 (Socher et al., 2013a), with black texts on a white background in $448 \times 448$ resolution. Each image is labeled as positive or negative based on the mood expressed in the text, and the number of images for both classes is nearly balanced. There are $6,920$ training and $1,821$ testing images.

**Language generation datasets.** We provide a detailed description of the 11 datasets used for language generation evaluation as follows.

- **IFEval** (Zhou et al., 2023) is a straightforward and easy-to-reproduce benchmark on instruction-following evaluation. It contains $541$ "verifiable instructions" such as "write in more than $400$ words" and "mention the keyword of AI at least 3 times". The dataset is publicly available at `https://huggingface.co/datasets/google/IFEval`.

- **GSM8K** (Cobbe et al., 2021) stands for Grade School Math $8$K, which contains $8,792$ high quality grade school math problems created by human writers. These problems take between $2$ and $8$ steps to solve, where the solutions primarily involve performing a sequence of basic arithmetic operations ($+ - \times \div$). There are $1,319$ test problems. The dataset is publicly available at `https://huggingface.co/datasets/openai/gsm8k`.

- **Multilingual MMLU**, **Multilingual ARC**, and **Multilingual Hellaswag** (Lai et al., 2023) are the ChatGPT-translated versions from English of three corresponding datasets, *i.e,* MMLU (Hendrycks et al., 2021), ARC (Clark et al., 2018), and Hellaswag (Zellers et al., 2019). Although there are $26$ languages, following (He et al., 2025), we evaluate the merge models on French, Spanish, German, and Russian. All of these datasets are organized as multiple-choice question-answering tasks, which focus on different types of knowledge. MMLU assesses the model's multi-task accuracy on a wide range of world knowledge and problem-solving ability. ARC challenges models on reasoning tasks, which comprise natural, grade-school science questions. Hellaswag provides commonsense natural language inference questions that are trivial for humans, but difficult for state-of-the-art models. These translated datasets are publicly provided as follows: Multilingual MMLU at `https://huggingface.co/datasets/alexandrainst/m_mmlu`, Multilingual ARC at `https://huggingface.co/datasets/alexandrainst/m_arc`, and Multilingual Hellaswag at `https://huggingface.co/datasets/alexandrainst/m_hellaswag`.

- **HumanEval+** and **MBPP+** (Liu et al., 2023) automatically augment the test-cases of the original HumanEval (Chen et al., 2021) and MBPP (Austin et al., 2021) datasets for code generation assessment. These benchmarks evaluate models' ability to synthesize programs from docstrings and natural language descriptions, respectively. HumanEval+ and MBPP+ provide 80x/35x more tests than the originals, with test splits consisting of $164$ and $378$ programming tasks, respectively. These augmented datasets are publicly provided as follows: HumanEval+ at `https://huggingface.co/datasets/evalplus/humanevalplus` and MBPP+ at `https://huggingface.co/datasets/evalplus/mbppplus`.

- **WildGuardTest** (Han et al., 2024) is a large-scale and carefully balanced multi-task safety moderation dataset. The dataset contains $1,725$ harmful and unharmful samples covering vanilla (direct) and adversarial prompts. However, we only evaluate the merge models' ability to detect harm using $754$ harmful samples. The dataset is publicly available at `https://huggingface.co/datasets/allenai/wildguardmix`.

- **HarmBench** (Mazeika et al., 2024) is a standardized evaluation framework for automated red teaming methods. HarmBench contains $400$ textual behaviors, split into $320$ behaviors for test and $80$ behaviors for validation. These behaviors are designed to violate laws or norms, such that LLMs should not exhibit them. Each behavior is further specified with two types of categorization: semantic and functional categories. Semantic category describes the type of harmful behavior, including cybercrime, copyright violations, and generating misinformation. Functional category describes properties of behaviors, which help measure LLM's robustness. The dataset is publicly available at `https://github.com/nouhadziri/safety-eval-fork/blob/main/evaluation/tasks/generation/harmbench/harmbench_behaviors_text_test.csv`.

- **DoAnythingNow** (Shen et al., 2024) contains jailbreak prompts (spanning from December 2022 to December 2023), which are exploited by malicious users to bypass the safeguards and elicit harmful content from LLMs. These prompts are collected from four prominent platforms commonly used for prompt sharing: Reddit, Discord, websites, and open-source datasets. Following He et al. (2025), we evaluate the merge models on a subset of 300 jailbreak prompts created from February 2023 to April 2023. The dataset is publicly available at `https://github.com/nouhadziri/safety-eval-fork/blob/main/evaluation/tasks/generation/do_anything_now/do_anything_now_jailbreak.jsonl`.

- **XSTest** (Röttger et al., 2024) evaluates whether models' safeguards are exaggerated. XSTest is inspired by a circumstance where models often struggle to balance helpfulness and harmlessness: a clearly safe prompt is even refused if it uses similar language to unsafe ones or mentions sensitive topics. XSTest contains 450 test prompts: 250 safe prompts and 200 unsafe prompts. The dataset is publicly available at `https://github.com/nouhadziri/safety-eval-fork/blob/main/evaluation/tasks/generation/xstest/exaggerated_safety.json`.

### B.2 MODELS

**Vision classification models.** We employ off-the-shelf fine-tuned checkpoints from the previous work (Tang et al., 2024a), covering three architectures of pre-trained CLIP model (Radford et al., 2021): ViT-B/32, ViT-B/16, and ViT-L/14. The indicators 32, 16, and 14 mean the size of patches in pixels that an input image is divided into. We only merge the vision encoding part of these architectures, while the text encoding part is kept unchanged. The number of parameters for the vision encoding part of these architectures is 87.5M, 85.8M, and 303M, respectively. Fine-tuned checkpoints on the standard 8-task benchmark are publicly provided as follows: ViT-B/32 at `https://huggingface.co/collections/tanganke/clip-vit-b-32-on-the-eight-image-classication-tasks`, ViT-B/16 at `https://huggingface.co/collections/tanganke/clip-vit-b-16-on-the-eight-image-classification-tasks`, and ViT-L/14 at `https://huggingface.co/collections/tanganke/clip-vit-l-14-on-the-eight-image-classification-tasks`.

**Language generation models.** We employ off-the-shelf fine-tuned checkpoints from the previous work (He et al., 2025), covering two Llama 3 variants (Grattafiori et al., 2024): Llama-3.2-3B and Llama-3.1-8B. For further details on these checkpoints, we refer readers to Section 3.3 of He et al. (2025). Fine-tuned checkpoints are publicly provided as follows: Llama-3.2-3B at `https://huggingface.co/collections/MergeBench/llama-32-3b-models` and Llama-3.1-8B at `https://huggingface.co/collections/MergeBench/llama-31-8b-models`.

### B.3 MERGING METHODS

The merging methods we employ for comparison are listed in three groups as follows:

1. *Data-Free Methods:*
   - **Model Soups** (Wortsman et al., 2022) is the most straightforward approach that simply takes the average of candidate models' parameters to produce a merge model.
   - **Task Arithmetic** (Ilharco et al., 2022) introduces a concept called "task vector", which is the difference between the fine-tuned model's parameters and the pre-trained parameters. A multi-tasking task vector is defined as the sum of those task vectors and is scaled by a coefficient before being added back to the pre-trained model's parameters to produce a merge model.
   - **TIES-Merging** (Yadav et al., 2023) proposes to trim the small values of task vectors, then resolve sign conflicts before adding back to the pre-trained parameters.
   - **TSV-M** (Gargiulo et al., 2025) compresses the task vectors using singular value decomposition (SVD) to reduce the interference between task vectors at the layer level before merging.

- **DOGE TA** (Wei et al., 2025) is a variant of Task Arithmetic, where DOGE, an iterative algorithm minimizing the gap between the merge model and the candidates while retaining the shared knowledge, is integrated.
- **Iso-C** and **Iso-CTS** (Marczak et al., 2025). The former applied SVD on the merge task vector to identify the directions amplified by multiple tasks, *i.e.,* common subspace. The latter further incorporates task-specific subspaces for retaining unique task features.

2. *Training-Free Methods:*

- **Fisher Merging** (Matena & Raffel, 2022) produces the merge models by taking the weighted average of candidate models, with the weighting factors determined by the Fisher information matrices.
- **RegMean** (Jin et al., 2022) proposes a closed-form solution for merging multiple linear layers, then applies this idea to the transformer models.

3. Test-Time Adaptation:

- **Layer-wise AdaMerging** (Yang et al., 2024b) adaptively learns the merging coefficients introduced by Task Arithmetic in the layer-wise or task-wise manner by using unsupervised entropy minimization on unlabeled test datasets.
- **DOGE AM** (Wei et al., 2025) is another variant of AdaMerging, where DOGE is integrated.

## C  EXPERIMENTAL DETAILS

### C.1  NORMALIZED ACCURACY METRIC

To avoid distortions caused by differences in value ranges, we report normalized accuracy for experiments on large-scale tasks. The normalized accuracy for each task is computed relative to the accuracy of its corresponding candidate model, then averaged over tasks as:

$$\text{Avg. Norm. Accuracy} = \frac{1}{K} \sum_{i=1}^{K} \frac{\text{acc}[f_M(\boldsymbol{X}_i)]}{\text{acc}[f_i(\boldsymbol{X}_i)]}. \tag{4}$$

### C.2  LANGUAGE TASKS' METRICS

We provide the task-specific metrics for merging evaluation on language tasks in Table 7.

| Domain | Dataset | Metric |
|---|---|---|
| Instruction following | IFEval (Zhou et al., 2023) | Prompt level accuracy |
| Mathematics | GSM8K (Cobbe et al., 2021) | Exact match (8-shot Chain-of-Thought) |
| Multilingual understanding (on French, Spanish, German, and Russian) | Multilingual MMLU (Lai et al., 2023) Multilingual ARC (Lai et al., 2023) Multilingual Hellaswag (Lai et al., 2023) | Accuracy Normalized accuracy Normalized accuracy |
| Coding | HumanEval+ (Liu et al., 2023) MBPP+ (Liu et al., 2023) | Pass@1 Pass@1 |
| Safety | WildGuardTest (Han et al., 2024) HarmBench (Mazeika et al., 2024) DoAnythingNow (Shen et al., 2024) XSTest (Röttger et al., 2024) | Refuse to answer Refuse to answer Refuse to answer Accuracy |

Table 7: Task-specific metrics for merging evaluation on language tasks.

## C.3 HYPERPARAMETERS

| Method | 0.10 | 0.15 | 0.20 | 0.25 | 0.30 | 0.35 | 0.40 | 0.45 | 0.50 | 0.55 | 0.60 | 0.65 | 0.70 | 0.75 | 0.80 | 0.85 | 0.90 | 0.95 | 1.00 |
|---|---|---|---|---|---|---|---|---|---|---|---|---|---|---|---|---|---|---|---|
| *ViT-B/32* | | | | | | | | | | | | | | | | | | | |
| RegMean | 76.7 | 78.4 | 79.7 | 80.7 | 81.1 | 81.9 | 82.7 | 83.3 | 83.7 | 84.2 | 84.6 | 84.9 | 85.8 | 85.9 | 86.6 | 86.9 | 87.6 | 87.6 | 3.8 |
| **RegMean++** | 78.5 | 80.0 | 81.4 | 82.1 | 83.2 | 83.6 | 84.5 | 85.0 | 85.8 | 86.2 | 86.9 | 87.4 | 87.8 | 88.0 | 88.9 | 89.4 | 89.8 | 90.1 | 5.0 |
| *ViT-B/16* | | | | | | | | | | | | | | | | | | | |
| RegMean | 80.7 | 82.0 | 82.9 | 83.6 | 84.1 | 84.9 | 85.7 | 85.8 | 86.6 | 87.0 | 87.5 | 88.0 | 88.4 | 88.6 | 89.0 | 89.1 | 89.6 | 90.3 | 4.1 |
| **RegMean++** | 82.3 | 83.8 | 84.9 | 85.6 | 86.2 | 86.9 | 87.2 | 87.8 | 88.1 | 88.5 | 88.9 | 89.4 | 90.0 | 90.2 | 90.7 | 91.0 | 91.3 | 91.6 | 4.7 |
| *ViT-L/14* | | | | | | | | | | | | | | | | | | | |
| RegMean | 87.1 | 88.0 | 88.7 | 89.4 | 89.7 | 90.2 | 90.6 | 91.1 | 91.5 | 91.7 | 92.2 | 92.4 | 92.5 | 92.8 | 93.1 | 93.4 | 93.7 | 94.0 | 4.8 |
| **RegMean++** | 88.0 | 89.2 | 89.8 | 90.6 | 91.0 | 91.6 | 91.9 | 92.2 | 92.5 | 92.6 | 93.0 | 93.3 | 93.5 | 93.7 | 94.0 | 94.5 | 94.7 | 94.8 | 4.6 |

Table 8: Performance of 8-task scenario merging on held-out validation sets when varying the scaling factor $\alpha$ for non-diagonal items of the inner-product matrices.

**RegMean and RegMean++.** Both require two hyperparameters: the number of samples per task and the scaling factor $\alpha$. As illustrated in main text Figure 7 and Figure 8, increasing the number of samples for each task gives a relatively small improvement on performance across architectures. *We choose to use 256 samples for calculating the task's inner-product matrices* with a batch size of 32 as default. For the scaling factor $\alpha$, Table 8 shows that a higher $\alpha$ delivers better performance. However, when $\alpha = 1.0$, *i.e.,* no scaling applied, the degradation happens and the overall accuracy is almost zeroed out. This phenomenon is consistent with the insight from Jin et al. (2022). Therefore, $\alpha = 0.95$ *is the optimal value*.

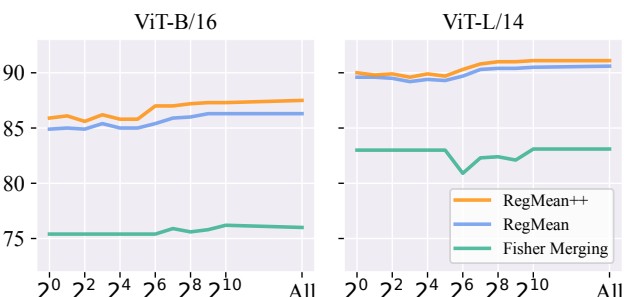

Figure 8: Effect of the amount of data needed for one candidate model when merging ViT-B/16 and ViT-L/14.

**Other methods.** We follow the suggestions on hyperparameters in the original works and set all of the hyperparameters below as default across experiments.

- For Task Arithmetic (Ilharco et al., 2022), the merging coefficient $\lambda = 0.3$ is used.

- For TIES-Merging (Yadav et al., 2023), top-20% highest-magnitude parameters are retained for each task vector, then these trimmed task vectors are merged with the merging coefficient $\lambda = 0.3$.

- For TSV-M (Gargiulo et al., 2025), the task scaling factor $\alpha = 1.0$ is used.

- For DOGE TA (Wei et al., 2025), the modification vector $\Delta$ is optimized on 400 iterations with a learning rate $1e-4$ via Adam optimizer (Kingma & Ba, 2014), the global magnitude of merging coefficient $\eta = 0.07$, the shared subspace basis size is set as the rank of the shared subspace divided by 6, and top-30% highest-magnitude parameters are retained for each task vector.

- For Iso-C (Marczak et al., 2025), the task scaling factor $\alpha$ is set to 1.30, 1.40, and 1.50 for ViT-B/32, ViT-B/16, and ViT-L/14, respectively.

- For Iso-CTS (Marczak et al., 2025), the task scaling factor $\alpha$ is set to 1.50, 1.60, and 1.90 for ViT-B/32, ViT-B/16, and ViT-L/14, respectively; the size of the common subspace is set as its rank multiplied by 0.8.

- For Fisher Merging (Matena & Raffel, 2022), the number of samples per task is 256 and the batch size is 32 for calculating the Fisher information matrix.

- For Layer-wise AdaMerging (Yang et al., 2024b), the Adam optimizer is used with a learning rate of $1e-3$ for updating the merging coefficients on $1,000$ iterations with the batch size of 16.

- For DOGE AM (Wei et al., 2025), its hyperparameters are the same as DOGE TA and Layer-wise AdaMerging.

## C.4 DETAILS FOR EXPERIMENTAL SETTINGS

**Order of tasks for sustainability evaluation.** We assess the merging performance by varying the number of tasks $n \in \{2, 4, 8, 12, 16, 20\}$. For every iteration $i$, we get first $n_i$ tasks from a fixed sequence of 20 tasks, perform merging, and then *evaluate the performance on the $n_i$ involved tasks only*. These experiments are conducted independently and re-executed from scratch for every iteration. The fixed sequence of 20 tasks is as follows:

SUN397, Stanford Cars, RESISC45, EuroSAT, SVHN, GTSRB, MNIST, DTD, Flowers102, PCAM, FER2013, OxfordIIITPet, STL10, CIFAR100, CIFAR10, Food101, FashionMNIST, EMNIST, KMNIST, RenderedSST2.

**Procedure for sequential merging.** We assess the merging performance in a scenario where tasks arrive sequentially. We choose to merge four tasks at a time. Specifically, we first merge four task-specific models to obtain an initial merge model. Then, this merge model is further merged with the new four task-specific models, using the ID data for calculating its merging statistics. That is, a mixture of task-specific datasets is constructed, where the number of samples used for each task is simply defined as 256 divided by the total number of tasks involved so far. This merging process is repeated until all of 20 task-specific models are merged. Right after every merge model is obtained, we *evaluate its performance on all 20 tasks*.

To determine the order of merging tasks, we generate a batch of different task combinations and choose five task sequences among them such that every non-overlapping group of four tasks in a task sequence does not exist in the other sequences. The five task sequences are as follows:

- PCAM, FER2013, OxfordIIITPet, RenderedSST2, GTSRB, FashionMNIST, SUN397, CIFAR100, EuroSAT, Stanford Cars, MNIST, STL10, DTD, Flowers102, CIFAR10, Food101, KMNIST, EMNIST, SVHN, RESISC45.

- CIFAR100, SUN397, EMNIST, EuroSAT, RESISC45, Food101, Flowers102, PCAM, RenderedSST2, Stanford Cars, CIFAR10, GTSRB, MNIST, DTD, KMNIST, FashionMNIST, STL10, SVHN, OxfordIIITPet, FER2013.

- EuroSAT, RenderedSST2, SUN397, FashionMNIST, Food101, KMNIST, OxfordIIITPet, DTD, PCAM, FER2013, Flowers102, MNIST, RESISC45, Stanford Cars, CIFAR10, STL10, GTSRB, EMNIST, SVHN, CIFAR100.

- EMNIST, RESISC45, MNIST, CIFAR10, FashionMNIST, SVHN, KMNIST, STL10, GTSRB, EuroSAT, SUN397, PCAM, Flowers102, FER2013, OxfordIIITPet, Food101, DTD, RenderedSST2, Stanford Cars, CIFAR100.

- GTSRB, Stanford Cars, SUN397, FashionMNIST, CIFAR10, EMNIST, SVHN, FER2013, OxfordIIITPet, Food101, MNIST, RenderedSST2, DTD, CIFAR100, Flowers102, PCAM, KMNIST, STL10, EuroSAT, RESISC45.

**Unseen tasks for evaluating the OOD generalization ability.** We report the performance over five held-out sets for OOD tasks: {MNIST, DTD}, {SVHN, GTSRB}, {RESISC45, EuroSAT}, {SUN397, Cars}, and {Cars, RESISC45}. Meanwhile, the remaining six tasks in the standard benchmark serve as ID tasks. These sets are chosen such that the overlapping rate between OOD sets is the least.

**Example of corrupted images for evaluating the robustness.** Figure 15 demonstrates seven corrupted variants for a clean image drawn from the Stanford Cars dataset.

**ImageNet sampling (OOD sampling).** Due to its massive volume, we randomly select 256 from the first $10,000$ samples of this database. These selected samples are used as proxy task-specific data for all of the candidate models.

**Class-imbalance sampling.** We simulate a data-limited scenario where only a single class from each task-specific dataset is utilized for merging. We execute five runs for each of the training-free methods, using different classes from the set $\{0, 1, 2, 3, 4\}$ for each run. For example, the first run involves class 0; meaning that for a task dataset, at most 256 samples labeled as 0 are randomly selected to serve as data for merging.

## C.5 HARDWARES

All of the experiments were conducted on either an A100 GPU with 40GB memory or an A40 GPU with 48GB memory.

## D ADDITIONAL RESULTS AND ANALYSIS

### D.1 PERFORMANCE ON THE 8-TASK BENCHMARK

We provide the performance comparison of RegMean++ and other methods on the 8-task benchmark for ViT-B/16 and ViT-L/14 in Table 11 and Table 12, respectively. Visualizations of representation bias for RegMean and RegMean++ on these models are shown in Figure 9 and Figure 10.

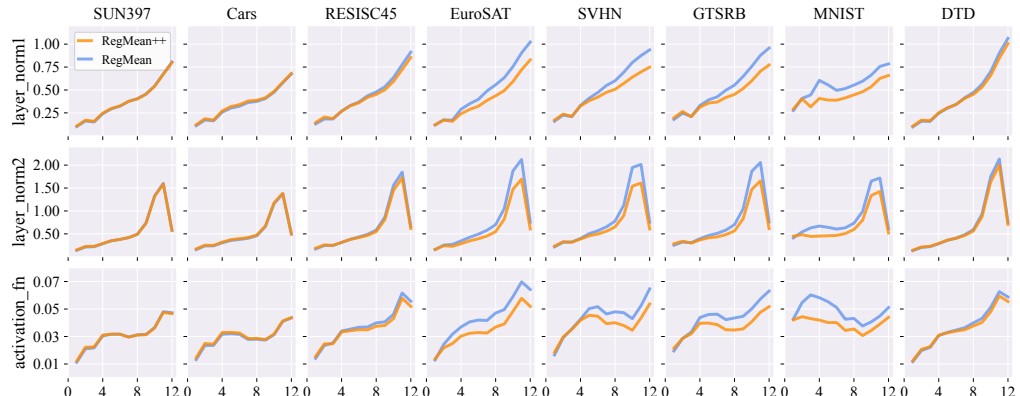

Figure 9: Representation bias in three non-linear components, namely two LayerNorms and a GELU activation, across transformer layers for RegMean and RegMean++ on 8-task merging with ViT-B/16 model.

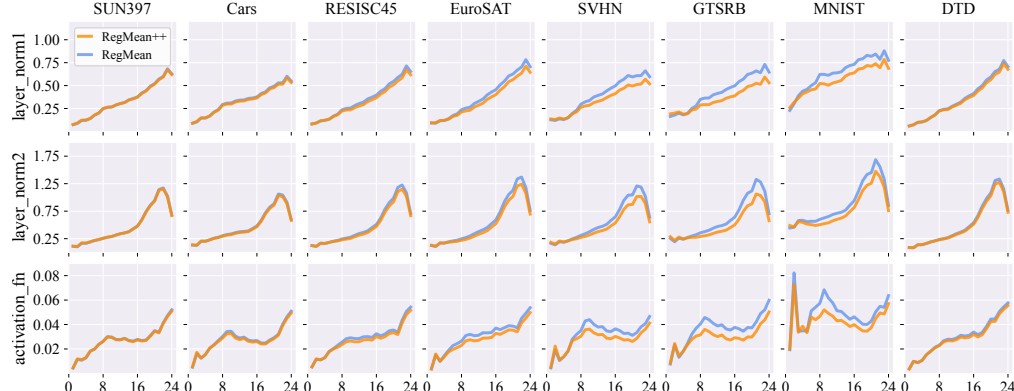

Figure 10: Representation bias in three non-linear components, namely two LayerNorms and a GELU activation, across transformer layers for RegMean and RegMean++ on 8-task merging with ViT-L/14 model.

## D.2 Sequential Merging

Along with the average performance on all 20 tasks shown in main text Figure 4, we additionally provide a more fine-grained analysis of RegMean and RegMean++ in Figure 11. Each entry in these heat maps visualizes the mean of average accuracy on a non-overlapping group of four tasks across five different task sequences. Along diagonals, which correspond to groups of current merging tasks, RegMean++'s performance matches or surpasses that of Reg-Mean. Furthermore, RegMean++ also exhibits a slightly enhanced ability to retain performance on earlier tasks, as indicated by the entries below those diagonals.

## D.3 Robustness Against Distribution Shifts

We provide the performance of Reg-Mean++ and other methods on the robustness analysis for ViT-B/16 and ViT-L/14 in Table 13 and Table 14, respectively.

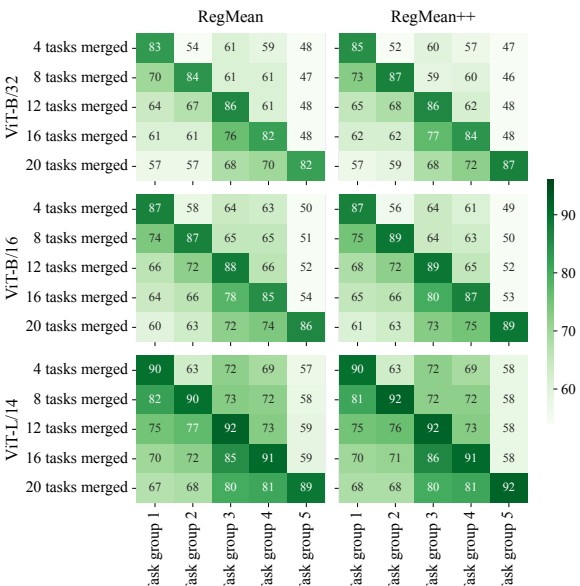

Figure 11: Details on sequential-merging performance comparison of RegMean and RegMean++ for three models. Each entry is the average accuracy on a group of four tasks, averaged over five runs.

## D.4 Merging in Different Spaces for Data-Free Methods

**Merging using middle and deep layers preserves overall performance or even outperforms all layers.** The results are reported in Table 9. For Task Arithmetic, TIES-Merging, TSV-M, and DOGE TA using both middle and deep layers achieves highly competitive performance, or even surpasses that of using all layers. The most noticeable performance differences can be observed on Task Arithmetic, where it consistently improves over full-layer merging by 6.6, 2.0, and 3.5 points for ViT-B/32, ViT-B/16, and ViT-L/14, respectively. Meanwhile, merging both middle and deep layers preserves from 98% to 99% performance for both Iso-C and Iso-CTS. Additionally, applying merging to only deep layers retains more than 92% overall performance, and is better than merging using middle or early layers. This trend can also be observed in layer-wise merging in Figure 12.

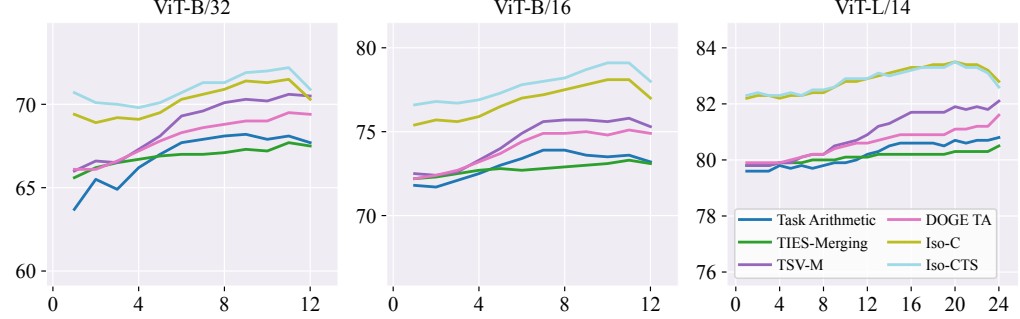

Figure 12: Layer-wise merging performance of data-free methods for ViT-B/32. ViT-B/16, and ViT-L/14. Results show average accuracy across eight tasks.

**MLP module linear layer merging is not always the best.** Component-specific merging analysis for data-free methods shows that using linear layers of the MLP modules still yields better performance than that of the attention heads, except for Task Arithmetic, as indicated in Table 9.

| Components | ViT-B/32 | | | | | | ViT-B/16 | | | | | | ViT-L/14 | | | | | |
|---|---|---|---|---|---|---|---|---|---|---|---|---|---|---|---|---|---|---|
| | TA | TIES | TSV-M | DOGE TA | Iso-C | Iso-CTS | TA | TIES | TSV-M | DOGE TA | Iso-C | Iso-CTS | TA | TIES | TSV-M | DOGE TA | Iso-C | Iso-CTS |
| All | 67.5 | 71.9 | 83.1 | 80.6 | 82.5 | 82.7 | 77.1 | 77.6 | 87.1 | 84.7 | 87.9 | 88.3 | 80.5 | 83.8 | 90.6 | 88.7 | 92.1 | 92.7 |
| Early | 60.5 | 65.8 | 66.6 | 66.5 | 70.7 | 71.7 | 70.7 | 72.7 | 73.2 | 72.8 | 77.3 | 78.3 | 76.6 | 79.8 | 80.1 | 80.4 | 83.9 | 84.8 |
| Middle | 70.1 | 69.0 | 74.9 | 72.8 | 74.7 | 75.0 | 76.4 | 74.1 | 80.2 | 79.0 | 81.5 | 81.6 | 80.9 | 81.7 | 85.9 | 84.4 | 87.3 | 87.9 |
| Deep | 72.1 | 70.6 | 78.5 | 76.3 | 77.0 | 76.9 | 76.4 | 75.5 | 82.6 | 81.1 | 83.4 | 83.9 | 84.1 | 82.7 | 88.1 | 86.4 | 88.8 | 88.7 |
| Middle & Deep | 74.1 | 73.1 | 83.4 | 81.0 | 81.6 | 81.0 | 79.1 | 77.3 | 87.0 | 84.9 | 87.0 | 87.0 | 84.0 | 84.2 | 90.9 | 88.6 | 91.4 | 91.8 |
| Attention heads | 71.1 | 69.2 | 76.0 | 75.0 | 73.6 | 73.9 | 76.6 | 74.7 | 80.7 | 80.9 | 79.5 | 79.7 | 83.6 | 82.2 | 87.1 | 85.8 | 87.0 | 88.1 |
| MLPs | 67.4 | 70.1 | 80.3 | 76.2 | 80.1 | 80.6 | 75.6 | 75.7 | 84.5 | 81.6 | 85.6 | 86.1 | 79.3 | 82.4 | 88.6 | 86.2 | 90.9 | 91.8 |

Table 9: Region-specific merging and component-specific merging performance of data-free methods for ViT-B/32, ViT-B/16, and ViT-L/14, where Task Arithmetic is denoted as TA and TIES-Merging is denoted as TIES. Results show average accuracy across eight tasks.

## D.5 REPRESENTATION BIAS

Figure 13 shows a comparison of the representation bias in the last layer when applying RegMean and RegMean++ on ViT-B/32, ViT-B/16, and ViT-L/14.

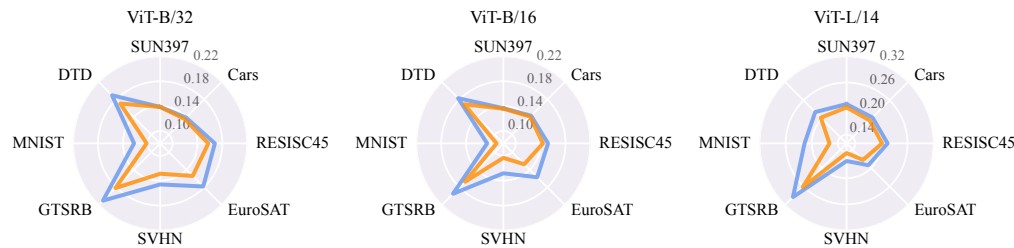

Figure 13: Representation bias in the last layer for ViT-B/32, ViT-B/16, and ViT-L/14 of RegMean++ (dark orange) and RegMean (cornflower blue).

## D.6 FULL RESULTS ON VARYING DATA CHARACTERISTICS ANALYSIS

In main text Table 5, we have provided the average accuracy across eight tasks. In Figure 14, we provide a detailed performance of data-free methods under different effects of data characteristics.

## D.7 COMPUTATIONAL REQUIREMENTS

We measure the computational time (in seconds) and peak GPU memory requirement (in GB) on 8-task merging for all algorithms and report the statistics in the Table 10. All of the statistics are measured on a single A100 GPU with 40GB of memory.

In terms of merging time, RegMean++ incurs minimal overhead compared to RegMean on ViT-B/32 models (34s vs. 31s). But the gap becomes more pronounced for merging larger models. On ViT-L/14, RegMean++ requires roughly 2x merging time compared to RegMean (171s vs. 84s) for additional forward passes over all of the data after every merge layer is obtained.

In terms of memory overhead, RegMean++ incurs no additional overhead compared to RegMean. Both require approximately 4GB and 13GB of GPU memory for merging ViT-B/32 and ViT-L/14 models, respectively. These memory requirements are comparable to those of data-free methods, cheaper than those of test-time adaptation methods, and even much cheaper than Fisher Merging.

Compared to re-training, it is worth noting that RegMean++ is dramatically faster and more memory-efficient. First, RegMean++ requires iteration through only 256 samples per task (0.35% to 6.81% of the task's training dataset) for operation. Hence, RegMean++ is dramatically faster compared to re-training, which requires iteration through the full datasets. Second, re-training requires peak memory usage comparable to Fisher Merging (He et al., 2025) for both forward and backward passes for gradient computation. RegMean++, on the other hand, does not need any backward pass. In our experiments on ViT-L/14 models, RegMean++ needs nearly 13GB GPU memory, which is far less than that of Fisher Merging with almost 35GB.

| Method | ViT-B/32 | | ViT-L/14 | |
|---|---|---|---|---|
| | Merging Time | GPU Memory | Merging Time | GPU Memory |
| *Data-Free Methods* | | | | |
| Model Soups | 0.06 | 3.26 | 0.09 | 11.30 |
| Task Arithmetic | 0.08 | 3.91 | 0.15 | 13.56 |
| TIES-Merging | 1.20 | 3.64 | 3.91 | 11.97 |
| TSV-M | 40.64 | 3.59 | 127.81 | 12.44 |
| DOGE TA | 128.59 | 4.76 | 365.57 | 16.17 |
| Iso-C | 4.29 | 3.92 | 12.91 | 13.57 |
| Iso-CTS | 62.92 | 3.92 | 179.59 | 13.57 |
| *Training-Free Methods* | | | | |
| Fisher Merging | 27.71 | 6.74 | 106.42 | 34.44 |
| RegMean | 31.30 | 3.76 | 84.09 | 12.69 |
| **RegMean++** (Ours) | 34.17 | 4.04 | 171.42 | 12.69 |
| *Test-Time Adaptation* | | | | |
| Layer-wise AdaMerging | 670.58 | 4.76 | 5195.43 | 27.62 |
| DOGE AM | 636.63 | 6.22 | 5452.13 | 27.51 |

Table 10: Computational requirements of different merging methods.

# E    LIMITATIONS AND FUTURE WORK

A primary limitation of RegMean++ is the increased computational overhead during the statistic-collection phase. Unlike RegMean, which can collect merging statistics from candidate models in parallel or using the pre-computed features, RegMean++ introduces a sequential dependency: the input features for the current layer $l$ depend on the outputs of the previous merge layer $l-1$. Although it makes the output representations of the merge model more aligned with those of the candidates, it necessitates one additional forward pass through the evolving merge model for each dataset. Consequently, this dynamic feature flow results in an approximate 2x increase in total merging time compared to the standard RegMean.

Due to the computational constraints, experiments are conducted with the largest models scaled to 8 billion parameters. This may risk overlooking interesting aspects of generalization. Thus, future work exploring RegMean++ at a higher model scale could be a promising direction.

# F    AI USAGE DECLARATION

AI tools were used for grammar checking, sentence rewriting for clarification, and figure and table formatting. All technical content and implementations were written by the authors.

| Method | SUN397 | Cars | RESISC45 | EuroSAT | SVHN | GTSRB | MNIST | DTD | Avg. |
|---|---|---|---|---|---|---|---|---|---|
| | | | | *Reference Results* | | | | | |
| Fine-tuned | 78.9 | 85.9 | 96.6 | 99.0 | 97.6 | 99.0 | 99.7 | 82.3 | 92.3 |
| | | | | *Data-Free Methods* | | | | | |
| Model Soups | 68.7 | 69.0 | 75.1 | 83.3 | 75.0 | 62.6 | 93.8 | 51.2 | 72.3 |
| Task Arithmetic | 65.9 | 68.3 | 75.5 | 84.5 | 88.9 | 82.0 | 98.1 | 54.0 | 77.1 |
| TIES-Merging | 70.7 | 71.2 | 79.9 | 87.5 | 83.3 | 76.3 | 96.4 | 55.5 | 77.6 |
| TSV-M | 73.1 | 80.7 | 89.7 | 96.2 | 94.1 | 94.1 | **99.1** | 69.7 | 87.1 |
| DOGE TA | 70.8 | 77.5 | 85.9 | 95.1 | 92.7 | 91.4 | 98.8 | 65.3 | 84.7 |
| Iso-C | 75.9 | 82.9 | 92.3 | 96.3 | 91.1 | 94.5 | 98.7 | 71.2 | 87.9 |
| Iso-CTS | **76.2** | **83.8** | **92.6** | 96.0 | 90.9 | 94.7 | 98.6 | 73.7 | **88.3** |
| | | | | *Training-Free Methods* | | | | | |
| Fisher Merging | 68.5 | 69.7 | 73.6 | 96.3 | 78.8 | 73.3 | 90.4 | 54.0 | 75.6 |
| RegMean | 72.6 | 78.8 | 89.2 | 96.3 | 94.9 | 90.0 | 98.8 | 67.9 | 86.0 |
| **RegMean++** (Ours) | 72.8 | 78.9 | 89.3 | 97.3 | 96.0 | 93.0 | **99.1** | 71.0 | 87.2 |
| | | | | *Test-Time Adaption Methods* | | | | | |
| Layer-wise AdaMerging | 70.7 | 79.8 | 86.5 | 93.4 | 93.7 | 95.6 | 98.1 | 62.7 | 85.1 |
| DOGE AM | 72.6 | 82.4 | 90.5 | 94.6 | 94.8 | **96.4** | 98.6 | **75.8** | 88.2 |

Table 11: Performance of all merging methods for ViT-B/16 measured on the 8-task benchmark. The **global best**, local best, and global runner-up are marked.

| Method | SUN397 | Cars | RESISC45 | EuroSAT | SVHN | GTSRB | MNIST | DTD | Avg. |
|---|---|---|---|---|---|---|---|---|---|
| | | | | *Reference Results* | | | | | |
| Fine-tuned | 82.8 | 92.9 | 97.4 | 99.2 | 97.9 | 99.2 | 99.8 | 85.5 | 94.3 |
| MTL | 79.0 | 89.3 | 94.5 | 98.4 | 96.4 | 98.1 | 99.4 | 83.7 | 92.4 |
| | | | | *Data-Free Methods* | | | | | |
| Model Soups | 72.5 | 81.5 | 82.3 | 88.5 | 81.6 | 74.0 | 96.6 | 61.8 | 79.9 |
| Task Arithmetic | 72.0 | 79.0 | 80.6 | 84.6 | 87.5 | 83.5 | 98.0 | 58.5 | 80.5 |
| TIES-Merging | 74.8 | 83.2 | 86.5 | 89.7 | 89.7 | 85.2 | 97.8 | 63.9 | 83.8 |
| TSV-M | 78.2 | 89.8 | 93.5 | 96.7 | 95.6 | 96.5 | 99.1 | 75.3 | 90.6 |
| DOGE TA | 76.6 | 87.5 | 91.3 | 96.0 | 94.4 | 93.5 | 98.9 | 71.3 | 88.7 |
| Iso-C | **80.7** | 91.5 | 95.3 | 97.2 | 95.1 | 97.8 | **99.1** | 80.3 | 92.1 |
| Iso-CTS | **80.7** | 92.2 | 95.9 | 97.5 | 95.7 | 98.4 | 99.2 | 82.1 | **92.7** |
| | | | | *Training-Free Methods* | | | | | |
| Fisher Merging | 70.9 | 78.8 | 83.0 | 94.7 | 84.9 | 94.9 | 91.1 | 61.0 | 82.4 |
| RegMean | 76.9 | 89.8 | 93.0 | **97.5** | 96.3 | 94.1 | 98.7 | 77.0 | 90.4 |
| **RegMean++** (Ours) | 77.2 | 89.6 | 92.8 | **97.5** | 96.9 | 96.3 | 99.2 | 78.4 | 91.0 |
| | | | | *Test-Time Adaption Methods* | | | | | |
| Layer-wise AdaMerging | 78.2 | 90.8 | 90.8 | 96.1 | 95.0 | 97.5 | 98.5 | 81.4 | 91.0 |
| DOGE AM | 79.6 | 91.8 | 94.2 | 96.8 | 96.3 | **98.6** | 98.9 | **83.8** | 92.5 |

Table 12: Performance of all merging methods for ViT-L/14 measured on the 8-task benchmark. The **global best**, local best, and global runner-up are marked.

| Method | Clean Test set | Corrupted Test set | | | | | | | |
|---|---|---|---|---|---|---|---|---|---|
| | | Motion | Impulse | Gaussian | Pixelate | Spatter | Contrast | JPEG | **Avg.** |
| | | *Data-Free Methods* | | | | | | | |
| Model Soups | 81.9 | 73.6 | 62.0 | 64.3 | 34.4 | 64.3 | 73.2 | 72.6 | 63.5 |
| Task Arithmetic | 84.0 | 75.8 | 64.1 | 65.9 | 35.4 | 66.5 | 75.2 | 74.6 | 65.3 |
| TIES-Merging | 78.4 | 69.2 | 57.5 | 59.7 | 30.3 | 60.7 | 68.7 | 68.6 | 59.2 |
| TSV-M | 92.3 | 86.0 | 74.6 | 73.3 | 42.0 | 77.5 | 84.1 | 84.8 | 74.6 |
| DOGE TA | 90.5 | 83.7 | 71.9 | 70.7 | 40.7 | 74.5 | 82.1 | 82.0 | 72.2 |
| Iso-C | 90.3 | 83.3 | 67.8 | 69.2 | 39.4 | 73.3 | 82.6 | 81.0 | 70.9 |
| Iso-CTS | 90.7 | 83.7 | 67.9 | 69.6 | 38.7 | 74.1 | 83.2 | 81.5 | 71.2 |
| | | *Training-Free Methods* | | | | | | | |
| Fisher Merging | 81.4 | 73.6 | 58.4 | 59.9 | 33.6 | 63.7 | 72.4 | 72.0 | 61.9 |
| RegMean | 92.6 | 86.2 | **75.6** | 73.3 | 41.9 | 77.7 | 84.7 | 84.8 | 74.9 |
| **RegMean++** (Ours) | **93.2** | 87.3 | 75.3 | 72.7 | 42.9 | 77.4 | 84.2 | 86.5 | 75.2 |
| | | *Test-Time Adaptation* | | | | | | | |
| Layer-wise AdaMerging | 90.9 | 84.6 | 71.7 | 73.0 | 42.9 | 75.7 | 83.3 | 83.6 | 73.5 |
| DOGE AM | 93.1 | **87.6** | 75.4 | **76.0** | **46.4** | 79.3 | 86.2 | 86.6 | **76.8** |

Table 13: Performance of merging methods for ViT-B/16 on corrupted test data. The **global best**, local best, and global runner-up are marked.

| Method | Clean Test set | Corrupted Test set | | | | | | | |
|---|---|---|---|---|---|---|---|---|---|
| | | Motion | Impulse | Gaussian | Pixelate | Spatter | Contrast | JPEG | **Avg.** |
| | | *Data-Free Methods* | | | | | | | |
| Model Soups | 89.7 | 82.3 | 71.7 | 72.3 | 37.9 | 75.1 | 81.2 | 80.8 | 71.6 |
| Task Arithmetic | 90.7 | 82.8 | 73.0 | 72.7 | 37.7 | 76.6 | 81.9 | 81.4 | 72.3 |
| TIES-Merging | 87.9 | 80.9 | 69.0 | 71.4 | 37.5 | 71.9 | 80.2 | 79.1 | 70.0 |
| TSV-M | 95.5 | 89.3 | 81.4 | 80.8 | 42.4 | 86.6 | 87.1 | 88.3 | 79.4 |
| DOGE TA | 94.4 | 88.5 | 79.0 | 79.8 | 44.1 | 83.8 | 87.0 | 87.0 | 78.5 |
| Iso-C | 95.7 | 90.5 | 79.5 | 81.8 | 45.9 | 86.1 | 89.1 | 88.5 | 80.2 |
| Iso-CTS | 96.2 | 90.8 | 81.0 | 82.4 | 45.3 | 87.4 | 89.5 | 89.0 | 80.8 |
| | | *Training-Free Methods* | | | | | | | |
| Fisher Merging | 89.2 | 81.9 | 73.7 | 72.5 | 40.2 | 77.6 | 80.8 | 81.0 | 72.5 |
| RegMean | 95.9 | 89.2 | 83.5 | 80.9 | 40.9 | 87.2 | 87.5 | 88.7 | 79.7 |
| **RegMean++** (Ours) | 96.1 | 89.8 | **83.7** | 80.4 | 41.5 | 87.4 | 87.5 | 89.4 | 79.9 |
| | | *Test-Time Adaptation* | | | | | | | |
| Layer-wise AdaMerging | 95.3 | 91.2 | 78.9 | 84.2 | 50.8 | **88.0** | 89.8 | 90.3 | 81.9 |
| DOGE AM | **96.4** | **91.6** | 80.8 | **85.5** | 50.6 | 86.6 | 90.8 | 91.3 | 82.5 |

Table 14: Performance of merging methods for ViT-L/14 on corrupted test data. The **global best**, local best, and global runner-up are marked.

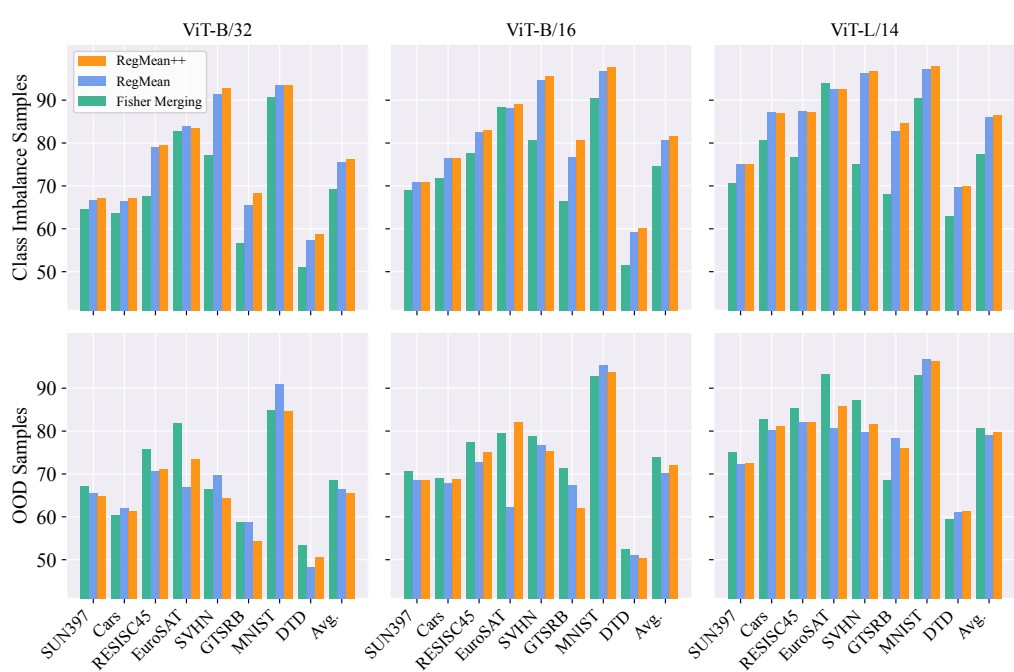

Figure 14: Accuracy of data-free methods when using class imbalance samples and OOD samples (ImageNet) for merging. For class imbalance, we report the mean of accuracy over five different classes.

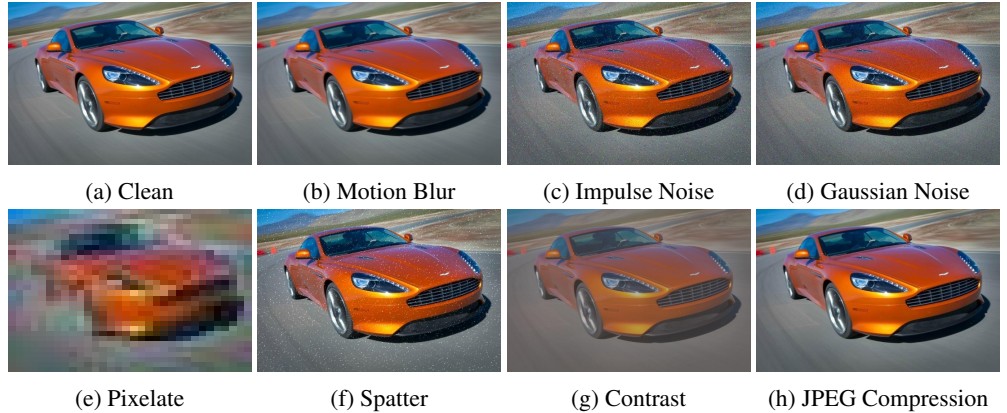

Figure 15: A visualization of a clean image and its corrupted versions with seven types of common noises (Hendrycks & Dietterich, 2019).

