# OpenReview forum: "RegMean++: Enhancing Effectiveness and Generalization of Regression Mean for Model Merging"
_ICLR.cc/2026/Conference — Submitted to ICLR 2026_

### Official Review · Reviewer_tzZH · 2025-10-28

**Soundness:** 3
**Presentation:** 3
**Contribution:** 2
**Rating:** 6
**Confidence:** 4

**Summary:**

This paper addresses the problem of model-merging: combining multiple task-specific models into a unified model without retraining. The authors engage with an existing method called RegMean (Regression Mean) which treats each linear layer in the merge model independently to minimize prediction error. They argue that RegMean neglects cross-layer and intra-layer dependencies (i.e., how features propagate through layers and how earlier layers affect later ones). This paper makes the following contributions:


* A new method called RegMean++ which extends RegMean by explicitly incorporating both intra-layer dependencies and cross-layer dependencies into the regression objective (i.e., modelling how layer outputs and later layers interact).


* Extensive empirical evaluation showing that RegMean++ outperforms RegMean across diverse settings: in-domain (ID) and out-of-domain (OOD) generalization, sequential merging (adding tasks one after another), large-scale tasks, and robustness under various distribution shifts.

* A demonstration that this methodology competes with or even beats recent advanced model-merging methods (beyond just the baseline RegMean) in some cases.

**Strengths:**

* The authors identify a meaningful shortcoming in RegMean: the layer-wise independence assumption ignores downstream effects and feature propagation. The articulation of this gap is clear.



* According to the submission summary, the method is evaluated across multiple axes (ID vs. OOD, sequential merging, large-scale tasks, robustness) and consistently outperforms the baseline (RegMean).

**Weaknesses:**

* While the regression‐mean merging paradigm is attractive, it implicitly assumes linear relationships between the candidate models and the merged model’s weights. The world of deep networks is highly nonlinear and features propagate in complex ways; how valid is the linear regression assumption in deep networks? Extra discussion needed.


* The paper claims to model intra‐ and cross‐layer dependencies, but it may be unclear exactly how much additional modelling is done (e.g., are inter‐layer weights estimated, correlation structures learned?) and how scalable that is to very deep architectures.


* It would strengthen the work to include analyses of when RegMean++ fails, e.g., tasks that are very dissimilar or when layer dependencies are minimal, or whether the benefit is marginal in some cases—and what trade‐offs exist.

**Questions:**

* Compared with baseline RegMean (which is relatively cheap), how much extra computation or memory cost does RegMean++ incur (both at merging time, and at inference time if relevant)? Are there any deployment concerns?

---

> ### Author Response · Authors · 2025-11-22
> **Official Comments from Authors (1/2)**
>
> We thank the reviewer for the comments. Please see our responses to your concerns below.
>
> ---
>
> > **Q**: While the regression‐mean merging techniques are reasonable, it implicitly assumes linear relationships between the candidate models and the merged model’s weights. However, in practice, the relations among different neural networks are highly nonlinear. Does the linear regression assumption still hold in deep networks?
>
> **A**: We thank the reviewer for this interesting question. We would like to discuss that, Regression-mean merging techniques (RegMean and RegMean++) solve a closed-form linear regression problem to find the merge weights. However, between these linear layers exist non-linear components such as GELU, ReLU, and LayerNorms.
>
> We believe the reviewer is implicitly asking about *inter-layer dependency*. If we merge layer $l$ using linear regression, the output of layer $l$ changes slightly (as the objective of RegMean and RegMean++). Because the deep net is non-linear, even a small change might potentially cause a large, unpredictable shift in the input distribution for layer $l+1$. We agree with the reviewer regarding the nonlinearity of these dependencies.
>
> However, we believe this question is actually a **critique of the original RegMean**, which is **exactly the motivation behind RegMean++**.
>
> Here, we would discuss that the original RegMean assumed independence (ignoring the non-linear propagation effects). RegMean++ specifically addresses the reviewer's concern by explicitly calculating the input features based on the **previous merge** layers (Algorithm 1, lines 4-6), rather than the **candidate previous** layers.
>
> We will add this as our motivation in the next revision.
>
> > **Q**: Compared with existing baseline approaches, it would be better to discuss how much more memory/computation cost does RegMean++ need, e.g., extra merging time? This may provide a more comprehensive evaluation.
>
> **A**: We measure the computational time (in seconds) and peak GPU memory requirement (in MiB) on 8-task merging for all algorithms and report the statistics in the table below.
>
> In terms of merging time, RegMean++ incurs minimal overhead compared to RegMean on ViT-B/32 models (34s vs. 31s). But the gap becomes more pronounced for merging larger models. On ViT-L/14, RegMean++ requires roughly 2x merging time compared to RegMean (171s vs. 84s) for additional forward passes over all of the data after every merge layer is obtained.
>
> In terms of memory overhead, RegMean++ incurs no additional overhead compared to RegMean. Both require approximately 4GB and 13GB of GPU memory for merging ViT-B/32 and ViT-L/14 models, respectively. These memory requirements are comparable to those of data-free methods, cheaper than those of test-time adaptation methods, and even much cheaper than Fisher Merging.
>
> Compared to re-training, it is worth noting that RegMean++ is dramatically faster and more memory-efficient. First, RegMean++ requires iteration through only 256 samples per task (0.35\% to 6.81\% of the task's training dataset) for operation. Hence, RegMean++ is dramatically faster compared to re-training, which requires iteration through the full datasets. Second, re-training requires peak memory usage comparable to Fisher Merging [1] for both forward and backward passes for gradient computation. RegMean++, on the other hand, does not need any backward pass. In our experiments on ViT-L/14 models, RegMean++ needs only 13GB GPU memory, which is far less than that of Fisher Merging with 35GB.
>
>
> | Method| ViT-B/32 Merging Time | ViT-B/32 GPU Memory | ViT-L/14 Merging Time | ViT-L/14 GPU Memory |
> |-|-|-|-|-|
> | Model Soups | 0.058787 | 3336.19 | 0.092278 | 11572.40 |
> | Task Arithmetic | 0.084564 | 4003.42 | 0.147474 | 13886.88 |
> | TIES-Merging | 1.201719 | 3723.18 | 3.912251 | 12259.68 |
> | TSV-M | 40.639873 | 3678.55 | 127.809463 | 12737.91 |
> | DOGE TA | 128.587673 | 4869.68 | 365.565765 | 16557.97 |
> | Iso-C | 4.288695 | 4012.42 | 12.907777 | 13893.46 |
> | Iso-CTS | 62.921379 | 4012.42 | 179.593353 | 13894.86 |
> | Fisher Merging | 27.713678 | 6905.90 | 106.419584 | 35270.44 |
> | RegMean | 31.299265 | 3855.25 | 84.094922 | 12996.90 |
> | RegMean++ (Ours) | 34.165437 | 4141.69 | 171.417743 | 12994.62 |
> | Layer-wise AdaMerging | 670.580244 | 4871.00 | 5195.433802 | 28287.00 |
> | DOGE AM | 636.628107 | 6372.12 | 5452.128822 | 28168.09 |
>
> *All of the statistics are measured on a single A100 GPU with 40GB of memory.*
>
> [1] He, Yifei, et al. "MergeBench: A Benchmark for Merging Domain-Specialized LLMs." arXiv preprint arXiv:2505.10833 (2025).

---

> ### Author Response · Authors · 2025-11-22
> **Official Comments from Authors (2/2)**
>
> > **Q**: The paper identify model intra‐ and cross‐layer dependencies, but it may be unclear that what is the performance that when scaling the method to very deep architectures. Does the proposed method still work in this case?
>
> **A**: We conducted additional experiments on Llama-3.2-3B and Llama-3.1-8B models with RegMean and RegMean++. Fine-tuned checkpoints are employed from [10].
>
> Following [10], we merged five different candidates, each covers one of the following domains: instruction following, mathematics, multilingual understanding, coding, and safety. Accordingly, the merged models are evaluated on 11 benchmarks reflecting these domains, instruction following: IFEval [1], mathematics: GSM8K [2], multilingual understanding: Multilingual MMLU [3], Multilingual ARC [3], Multilingual Hellaswag [3], coding: Humaneval+ [4], MBPP+ [5], and safety: WildGuardTest [6], HarmBench [7], DoAnythingNow [8], XSTest [9].
>
> The two tables below show the results of merging candidates from Llama-3.2-3B and Llama-3.1-8B models. We perform grid search over different scaling factors $\alpha \in \{ 0.1, 0.3, 0.5, 0.7, 0.9 \}$, and report the best result based on the averaged multi-domain performance. For calculating the inner-product matrices, we use 256 samples for each domain with max sequence length of 2048, and batch size of 1. We find that RegMean++ outperforms RegMean when merging candidates of Llama-3.1-8B, while it lags behind RegMean when merging candidates of Llama-3.2-3B. However, RegMean++ slightly underperforms RegMean in the instruction following task. Overall, these results validate the versatility and effectiveness of RegMean++ as it is not confined to vision classification tasks; it can generally be applied to text generation tasks.
>
> *Llama-3.2-3B:*
>
> |Method| Instruction following |Math|Multilingual|Coding|Safety|Avg.|
> |-|-|-|-|-|-|-|
> |RegMean|8.3|35.5| 47.3| 39.2| 39.8| 34.0|
> |RegMean++|6.8|35.1| 47.4| 37.0| 38.5| 33.0|
>
> *Llama-3.1-8B:*
>
> |Method|Instruction following|Math|Multilingual|Coding|Safety|Avg.|
> |-|-|-|-|-|-|-|
> |RegMean|26.6|63.2|49.0|48.3|36.9|44.8|
> |RegMean++|11.1|65.8|53.1|52.3|46.3|45.7|
>
> > **Q**: It would be better to include analysis and discussions on the theoretical conditions that when RegMean++ would or would not work.
>
> **A**: In the theoretical perspective, the scaling factor $0 \le \alpha \le 1$ for non-diagonal items of the inner-product matrices plays an important role in the closed-form merging solution (Eqn. 2). This scaling factor $\alpha$ behaves as a regularization that encourages the merge weights to be closer to the candidate's weights.
>
> As discussed in Appendix C.2, L844-L851, and described in Appendix Figure 6, higher $\alpha$ delivers better merging performance. However, when $\alpha = 1.0$, *i.e.,* no scaling applied, the degradation happens and the merging performance is almost zeroed out. This phenomenon is consistent with the insight from [11].
>
> [1] Zhou, Jeffrey, et al. "Instruction-following evaluation for large language models." arXiv preprint arXiv:2311.07911 (2023).
>
> [2] Cobbe, Karl, et al. "Training verifiers to solve math word problems." arXiv preprint arXiv:2110.14168 (2021).
>
> [3] Lai, Viet, et al. "Okapi: Instruction-tuned large language models in multiple languages with reinforcement learning from human feedback." EMNLP 2023.
>
> [4] Chen, Mark. "Evaluating large language models trained on code." arXiv preprint arXiv:2107.03374 (2021).
>
> [5] Austin, Jacob, et al. "Program synthesis with large language models." arXiv preprint arXiv:2108.07732 (2021).
>
> [6] Han, Seungju, et al. "Wildguard: Open one-stop moderation tools for safety risks, jailbreaks, and refusals of llms." NeurIPS 2024.
>
> [7] Mazeika, Mantas, et al. "HarmBench: A Standardized Evaluation Framework for Automated Red Teaming and Robust Refusal." ICML 2024.
>
> [8] Shen, Xinyue, et al. "" do anything now": Characterizing and evaluating in-the-wild jailbreak prompts on large language models." CCS 2024.
>
> [9] Röttger, Paul, et al. "Xstest: A test suite for identifying exaggerated safety behaviours in large language models." NAACL 2024.
>
> [10] He, Yifei, et al. "MergeBench: A Benchmark for Merging Domain-Specialized LLMs." arXiv preprint arXiv:2505.10833 (2025).
>
> [11] Jin, Xisen, et al. "Dataless Knowledge Fusion by Merging Weights of Language Models." ICLR 2023.
>
> ---
>
> We hope these address your concerns. If any questions remain, we are happy to discuss further!

---

### Official Review · Reviewer_uBzW · 2025-10-29

**Soundness:** 3
**Presentation:** 3
**Contribution:** 2
**Rating:** 4
**Confidence:** 4

**Summary:**

This paper proposes a method called RegMean++ for multi-task model merging, which is built upon the previous RegMean work.

**Strengths:**

- This paper points out that RegMean considers only single-layer information during merging, while ignoring the role of cross-layer information flow.
- The paper compares the proposed method with existing approaches in terms of accuracy, out-of-distribution generalization, and performance under distribution shift scenarios.

**Weaknesses:**

- This paper is an improvement based on RegMean, and the main difference lies in the input features; thus, its novelty is limited.
- The paper only validates the method on ViT architectures and simple image classification tasks, lacking verification on LLMs and text generation tasks.
- The experiments do not provide a comparison of time costs among different model merging methods.
- In Table 2, it is unclear why the proposed method outperforms RegMean on OOD data; this conclusion lacks deeper analysis or theoretical explanation.

**Questions:**

In Table 1, the original paper reports 85.86 for TSV-M and 86.3 for ISO-C, while this paper reports 83.1 and 82.5, respectively. What causes this discrepancy in the reported results?

---

> ### Author Response · Authors · 2025-11-22
> **Official Comments from Authors (1/3)**
>
> We thank the reviewer for the comments. Please see our responses to your concerns below.
>
> ---
>
> > **Q**: This paper is an improvement based on RegMean, and the main difference lies in the input features; thus, its novelty is limited.
>
> **A**:
> We would like to argue that simply changing the feature flow cannot limit the novelty of our work. First, given the nature of the transformer architecture itself, which comprises layers stacked on top of each other, the way the features in the earlier layers propagate through its layers is non-trivial. However, RegMean fundamentally ignores this notion in the merge model and independently applies its closed-form solution to the linear layers of the transformer layers. Second, through extensive experiments, we show that RegMean++ consistently outperforms RegMean with reduced representation bias across tasks, validating the importance of this feature flow. As described in Section 5.8, by incorporating the intra- and cross-layer dependencies of the merge model into the merging process, RegMean++'s merge models better capture the behaviours of the candidate models.
>
> *With RegMean++, a simple yet effective alternative to RegMean, we make three contributions*:
>
> - Through extensive experiments, we prove that RegMean++ consistently improves in-domain performance, out-of-domain generalization, robustness, and sustainability to sequential merging or large-scale tasks with less representation bias.
>
> - Our layer-wise analysis reveals two critical findings that shed light on the efficient merging schema. First, for all existing merging methods, merging linear layers in only the middle and the deep transformer layers preserves over 98\% of the performance compared to using all. Second, merging linear layers in MLP modules consistently outperforms using those in the attention heads.
>
> - Benchmarking against eleven advanced model merging methods shows that RegMean++ achieves competitive or state-of-the-art performance. These results validate the effectiveness of our proposed algorithm.
>
> > **Q**: The experiments do not provide a comparison of time costs among different model merging methods.
>
> **A**: We measure the computational time (in seconds) and peak GPU memory requirement (in MiB) on 8-task merging for all algorithms and report the statistics in the table below.
>
> In terms of merging time, RegMean++ incurs minimal overhead compared to RegMean on ViT-B/32 models (34s vs. 31s). But the gap becomes more pronounced for merging larger models. On ViT-L/14, RegMean++ requires roughly 2x merging time compared to RegMean (171s vs. 84s) for additional forward passes over all of the data after every merge layer is obtained.
>
> In terms of memory overhead, RegMean++ incurs no additional overhead compared to RegMean. Both require approximately 4GB and 13GB of GPU memory for merging ViT-B/32 and ViT-L/14 models, respectively. These memory requirements are comparable to those of data-free methods, cheaper than those of test-time adaptation methods, and even much cheaper than Fisher Merging.
>
> Compared to re-training, it is worth noting that RegMean++ is dramatically faster and more memory-efficient. First, RegMean++ requires iteration through only 256 samples per task (0.35\% to 6.81\% of the task's training dataset) for operation. Hence, RegMean++ is dramatically faster compared to re-training, which requires iteration through the full datasets. Second, re-training requires peak memory usage comparable to Fisher Merging [1] for both forward and backward passes for gradient computation. RegMean++, on the other hand, does not need any backward pass. In our experiments on ViT-L/14 models, RegMean++ needs only 13GB GPU memory, which is far less than that of Fisher Merging with 35GB.
>
>
> | Method| ViT-B/32 Merging Time | ViT-B/32 GPU Memory | ViT-L/14 Merging Time | ViT-L/14 GPU Memory |
> |-|-|-|-|-|
> | Model Soups | 0.058787 | 3336.19 | 0.092278 | 11572.40 |
> | Task Arithmetic | 0.084564 | 4003.42 | 0.147474 | 13886.88 |
> | TIES-Merging | 1.201719 | 3723.18 | 3.912251 | 12259.68 |
> | TSV-M | 40.639873 | 3678.55 | 127.809463 | 12737.91 |
> | DOGE TA | 128.587673 | 4869.68 | 365.565765 | 16557.97 |
> | Iso-C | 4.288695 | 4012.42 | 12.907777 | 13893.46 |
> | Iso-CTS | 62.921379 | 4012.42 | 179.593353 | 13894.86 |
> | Fisher Merging | 27.713678 | 6905.90 | 106.419584 | 35270.44 |
> | RegMean | 31.299265 | 3855.25 | 84.094922 | 12996.90 |
> | RegMean++ (Ours) | 34.165437 | 4141.69 | 171.417743 | 12994.62 |
> | Layer-wise AdaMerging | 670.580244 | 4871.00 | 5195.433802 | 28287.00 |
> | DOGE AM | 636.628107 | 6372.12 | 5452.128822 | 28168.09 |
>
> *All of the statistics are measured on a single A100 GPU with 40GB of memory.*
>
> [1] He, Yifei, et al. "MergeBench: A Benchmark for Merging Domain-Specialized LLMs." arXiv preprint arXiv:2505.10833 (2025).

---

> > ### Author Response · Authors · 2025-11-22
> > **Official Comments from Authors (2/3)**
> >
> > > **Q**: The paper only validates the method on ViT architectures and simple image classification tasks, lacking verification on LLMs and text generation tasks.
> >
> > **A**: We conducted additional experiments on Llama-3.2-3B and Llama-3.1-8B models with RegMean and RegMean++. Fine-tuned checkpoints are employed from [10].
> >
> > Following [10], we merged five different candidates, each covers one of the following domains: instruction following, mathematics, multilingual understanding, coding, and safety. Accordingly, the merged models are evaluated on 11 benchmarks reflecting these domains, instruction following: IFEval [1], mathematics: GSM8K [2], multilingual understanding: Multilingual MMLU [3], Multilingual ARC [3], Multilingual Hellaswag [3], coding: Humaneval+ [4], MBPP+ [5], and safety: WildGuardTest [6], HarmBench [7], DoAnythingNow [8], XSTest [9].
> >
> > The two tables below show the results of merging candidates from Llama-3.2-3B and Llama-3.1-8B models. We perform grid search over different scaling factors $\alpha \in \{ 0.1, 0.3, 0.5, 0.7, 0.9 \}$, and report the best result based on the averaged multi-domain performance. For calculating the inner-product matrices, we use 256 samples for each domain with max sequence length of 2048, and batch size of 1. We find that RegMean++ outperforms RegMean when merging candidates of Llama-3.1-8B, while it lags behind RegMean when merging candidates of Llama-3.2-3B. However, RegMean++ slightly underperforms RegMean in the instruction following task. Overall, these results validate the versatility and effectiveness of RegMean++ as it is not confined to vision classification tasks; it can generally be applied to text generation tasks.
> >
> > *Llama-3.2-3B:*
> >
> > |Method| Instruction following |Math|Multilingual|Coding|Safety|Avg.|
> > |-|-|-|-|-|-|-|
> > |RegMean|8.3|35.5| 47.3| 39.2| 39.8| 34.0|
> > |RegMean++|6.8|35.1| 47.4| 37.0| 38.5| 33.0|
> >
> > *Llama-3.1-8B:*
> >
> > |Method|Instruction following|Math|Multilingual|Coding|Safety|Avg.|
> > |-|-|-|-|-|-|-|
> > |RegMean|26.6|63.2|49.0|48.3|36.9|44.8|
> > |RegMean++|11.1|65.8|53.1|52.3|46.3|45.7|
> >
> > [1] Zhou, Jeffrey, et al. "Instruction-following evaluation for large language models." arXiv preprint arXiv:2311.07911 (2023).
> >
> > [2] Cobbe, Karl, et al. "Training verifiers to solve math word problems." arXiv preprint arXiv:2110.14168 (2021).
> >
> > [3] Lai, Viet, et al. "Okapi: Instruction-tuned large language models in multiple languages with reinforcement learning from human feedback." EMNLP 2023.
> >
> > [4] Chen, Mark. "Evaluating large language models trained on code." arXiv preprint arXiv:2107.03374 (2021).
> >
> > [5] Austin, Jacob, et al. "Program synthesis with large language models." arXiv preprint arXiv:2108.07732 (2021).
> >
> > [6] Han, Seungju, et al. "Wildguard: Open one-stop moderation tools for safety risks, jailbreaks, and refusals of llms." NeurIPS 2024.
> >
> > [7] Mazeika, Mantas, et al. "HarmBench: A Standardized Evaluation Framework for Automated Red Teaming and Robust Refusal." ICML 2024.
> >
> > [8] Shen, Xinyue, et al. "" do anything now": Characterizing and evaluating in-the-wild jailbreak prompts on large language models." CCS 2024.
> >
> > [9] Röttger, Paul, et al. "Xstest: A test suite for identifying exaggerated safety behaviours in large language models." NAACL 2024.
> >
> > [10] He, Yifei, et al. "MergeBench: A Benchmark for Merging Domain-Specialized LLMs." arXiv preprint arXiv:2505.10833 (2025).

---

> ### Author Response · Authors · 2025-11-22
> **Official Comments from Authors (3/3)**
>
> > **Q**: In Table 2, it is unclear why the proposed method outperforms RegMean on OOD data; this conclusion lacks deeper analysis or theoretical explanation.
>
> **A**: We address this concern as follows. The iterative refinement of the merge activations implicitly behaves like a structural regularizer, which helps the merge model learns toward cross-candidate patterns.
>
> > **Q**: In Table 1, the original paper reports 85.86 for TSV-M and 86.3 for ISO-C, while this paper reports 83.1 and 82.5, respectively. What causes this discrepancy in the reported results?
>
> **A**: This performance difference is due to the difference in checkpoints employed. We used the sets of checkpoints released by FusionBench [3], which are different from [1, 2].
>
> Following this suggestion, we reproduced TSV-M [2], Iso-C, Iso-CTS [1], RegMean, and our RegMean++ on the set of checkpoints provided by [1, 2] under the same setup described in the paper. The reproduced performance for the 8-task merging, as shown in the table below, matches the one reported in the respective papers.
>
> |Method|ViT-B/32 Reproduced|ViT-B/32 Reported [1, 2]|ViT-B/16 Reproduced|ViT-B/16 Reported [1, 2]|ViT-L/14 Reproduced|ViT-L/14 Reported [1, 2]|
> |-|-|-|-|-|-|-|
> |TSV-M| 85.66| 85.86| 88.97| 89.01| 92.94| 92.98|
> |Iso-C| 86.25| 86.30| 90.46| 90.60| 94.33| 94.20|
> |Iso-CTS| 86.37| 86.20| 91.16|91.10| 94.78| 94.70|
> |RegMean| 80.47| -| 83.35|-| 88.07|-|
> |RegMean++| 82.62| -| 85.41|-| 89.99| -|
>
> [1] Marczak, Daniel, et al. "No task left behind: Isotropic model merging with common and task-specific subspaces." ICML 2025.
>
> [2] Gargiulo, Antonio Andrea, et al. "Task singular vectors: Reducing task interference in model merging." CVPR 2025.
>
> [3] Tang, Anke, et al. "Fusionbench: A comprehensive benchmark of deep model fusion." arXiv preprint arXiv:2406.03280 (2024).
>
> ---
>
> We hope these address your concerns. If any questions remain, we are happy to discuss further!

---

### Official Review · Reviewer_sT1A · 2025-10-31

**Soundness:** 1
**Presentation:** 1
**Contribution:** 1
**Rating:** 0
**Confidence:** 4

**Summary:**

This paper proposes an improvement over RegMean. However, it lacks clear motivation, presents limited novelty, and provides an insufficiently explained methodology.

**Strengths:**

Extensive and well-controlled empirical campaign: Main benchmark covers 8 diverse image tasks, 3 model sizes, and 11 baselines. Sustainability under scale tested up to 20 tasks and sequential arrival. Robustness evaluated on seven corruption types and class-imbalanced or ImageNet-OOD samples.

**Weaknesses:**

1)This paper lacks clear motivation, presents limited novelty, and provides an insufficiently explained methodology.
2)No theoretical grounding for the “corrected” statistics: The manuscript claims RegMean++ “incorporates intra- and cross-layer dependencies” but offers no proof or even informal argument that using merged-model activations minimises a meaningful objective that couples layers (Sec. 3.1–3.2). Consequently, convergence, optimality, or error bounds with respect to the true multi-task risk are absent.
3)Computational cost brushed aside: RegMean++ needs an extra forward pass through the growing merged model for every layer to collect statistics (Algorithm 1, line 5). No wall-clock or FLOP comparison is given; the abstract claims “no computational overhead of re-training” but omits this non-trivial overhead relative to RegMean.
4)Scalability to larger models unaddressed: All experiments use ViT-B/32, B/16, L/14 (≤303 M params). The limitation section itself flags billion-scale models as future work, so current evidence does not support generalisability to LLM or large multimodal scenarios.
Code and checkpoints are not yet released, which limits immediate reproducibility.

**Questions:**

What is the motivation of this paper, and what are its main contributions?

---

> ### Author Response · Authors · 2025-11-22
> **Official Comments from Authors (1/3)**
>
> We thank the reviewer for the comments. Please see our responses to your concerns below.
>
> ---
>
> > **Q**: 1)This paper lacks clear motivation, presents limited novelty, and provides an insufficiently explained methodology.
>
> > **Q**: What is the motivation of this paper, and what are its main contributions?
>
> **A**:  *Motivation and Novelty*: We would like to clarify that: RegMean++ explicitly incorporates the information flow within the ongoing merge model into the merging objective, which is a missing recipe of the existing closed-form merging method, *i.e.,* RegMean. Given the nature of transformer architecture, which comprises layers stacked on top of each other, the way the information flows through layers is non-trivial for maintaining good representations that influence the final predictions. This motivated us to propose RegMean++.
>
> The incorporation of this information flow consistently improves in-domain performance, out-of-domain generalization, robustness, and sustainability to sequential merging or large-scale tasks over the existing closed-form merging method. With less representation bias compared to RegMean, RegMean++'s merge models are beter aligned with the candidates.
>
>
> *Contributions*: Our three contributions are well detailed in Section 1, L070-L087:
>
> - We introduce RegMean++, which incorporates both intra- and cross-layer dependencies of the merge model's layers into the RegMean merging objective. Through extensive experiments, we prove that RegMean++ consistently improves in-domain performance, out-of-domain generalization, robustness, and sustainability to sequential merging or large-scale tasks with less representation bias.
>
> - Our layer-wise analysis reveals two critical findings that shed light on the efficient merging schema. First, for all existing merging methods, merging linear layers in only the middle and the deep transformer layers preserves over 98\% of the performance compared to using all. Second, merging linear layers in MLP modules consistently outperforms using those in the attention heads.
>
> - Benchmarking against eleven advanced model merging methods shows that RegMean++ achieves competitive or state-of-the-art performance. These results validate the effectiveness of our proposed algorithm.
>
> *Methodology explanation*: We would like to argue that RegMean++'s methodology explanation is **already described** clearly in Section 3.2. Its pseudocode is provided in Algorithm 1, and a comparison between RegMean and RegMean++ is further illustrated in Figure 1.
>
> In Section 3.1, L166-L170, we clearly point out the drawback of RegMean: it overlooks how the information flows in the merge model. Finally, in Section 3.2, we describe how RegMean++ solves this by actively recalculating the merging statistics of the merge model after its layer is obtained.
>
> > **Q**: 2)No theoretical grounding for the “corrected” statistics: The manuscript claims RegMean++ “incorporates intra- and cross-layer dependencies” but offers no proof or even informal argument that using merged-model activations minimises a meaningful objective that couples layers (Sec. 3.1–3.2). Consequently, convergence, optimality, or error bounds with respect to the true multi-task risk are absent.
>
> **A**: We would like to clarify the intra- and cross-layer dependencies as follows. By actively recomputing the merging statistics of a merge layer via additional forward passes through its input features, the intra-layer dependency is captured. The output features are then used to compute the merging statistics for the next layers. This feature flow forms the cross-layer dependency.
>
> RegMean++ is built upon RegMean, whose closed-form solution is theoretically grounded. As an empirical work, we show the effectiveness of RegMean++ through extensive experimental analyses. We show that RegMean++ consistently outperforms RegMean with reduced representation bias across tasks, validating the importance of the feature flow. As described in Section 5.8, by incorporating the intra- and cross-layer dependencies of the merge model into the merging process, RegMean++'s merge models better capture the behaviours of the candidate models.

---

> > ### Author Response · Authors · 2025-11-22
> > **Official Comments from Authors (2/3)**
> >
> > > **Q**: 3)Computational cost brushed aside: RegMean++ needs an extra forward pass through the growing merged model for every layer to collect statistics (Algorithm 1, line 5). No wall-clock or FLOP comparison is given; the abstract claims “no computational overhead of re-training” but omits this non-trivial overhead relative to RegMean.
> >
> > **A**:  We measure the computational time (in seconds) and peak GPU memory requirement (in MiB) on 8-task merging for all algorithms and report the statistics in the table below.
> >
> > In terms of merging time, RegMean++ incurs minimal overhead compared to RegMean on ViT-B/32 models (34s vs. 31s). But the gap becomes more pronounced for merging larger models. On ViT-L/14, RegMean++ requires roughly 2x merging time compared to RegMean (171s vs. 84s) for additional forward passes over all of the data after every merge layer is obtained.
> >
> > In terms of memory overhead, RegMean++ incurs no additional overhead compared to RegMean. Both require approximately 4GB and 13GB of GPU memory for merging ViT-B/32 and ViT-L/14 models, respectively. These memory requirements are comparable to those of data-free methods, cheaper than those of test-time adaptation methods, and even much cheaper than Fisher Merging.
> >
> > Compared to re-training, it is worth noting that RegMean++ is dramatically faster and more memory-efficient. First, RegMean++ requires iteration through only 256 samples per task (0.35\% to 6.81\% of the task's training dataset) for operation. Hence, RegMean++ is dramatically faster compared to re-training, which requires iteration through the full datasets. Second, re-training requires peak memory usage comparable to Fisher Merging [1] for both forward and backward passes for gradient computation. RegMean++, on the other hand, does not need any backward pass. In our experiments on ViT-L/14 models, RegMean++ needs only 13GB GPU memory, which is far less than that of Fisher Merging with 35GB. These validate our claim.
> >
> >
> > | Method                | ViT-B/32 Merging Time | ViT-B/32 GPU Memory | ViT-L/14 Merging Time | ViT-L/14 GPU Memory |
> > |-----------------------|-----------------------|---------------------|-----------------------|---------------------|
> > | Model Soups           | 0.058787              | 3336.19             | 0.092278              | 11572.40            |
> > | Task Arithmetic       | 0.084564              | 4003.42             | 0.147474              | 13886.88            |
> > | TIES-Merging          | 1.201719              | 3723.18             | 3.912251              | 12259.68            |
> > | TSV-M                 | 40.639873             | 3678.55             | 127.809463            | 12737.91            |
> > | DOGE TA               | 128.587673            | 4869.68             | 365.565765            | 16557.97            |
> > | Iso-C                 | 4.288695              | 4012.42             | 12.907777             | 13893.46            |
> > | Iso-CTS               | 62.921379             | 4012.42             | 179.593353            | 13894.86            |
> > | Fisher Merging        | 27.713678             | 6905.90             | 106.419584            | 35270.44            |
> > | RegMean               | 31.299265             | 3855.25             | 84.094922             | 12996.90            |
> > | RegMean++ (Ours)      | 34.165437             | 4141.69             | 171.417743            | 12994.62            |
> > | Layer-wise AdaMerging | 670.580244            | 4871.00             | 5195.433802           | 28287.00            |
> > | DOGE AM               | 636.628107            | 6372.12             | 5452.128822           | 28168.09            |
> >
> > *All of the statistics are measured on a single A100 GPU with 40GB of memory.*
> >
> > [1] He, Yifei, et al. "MergeBench: A Benchmark for Merging Domain-Specialized LLMs." arXiv preprint arXiv:2505.10833 (2025).

---

> ### Author Response · Authors · 2025-11-22
> **Official Comments from Authors (3/3)**
>
> > **Q**: 4)Scalability to larger models unaddressed: All experiments use ViT-B/32, B/16, L/14 (≤303 M params). The limitation section itself flags billion-scale models as future work, so current evidence does not support generalisability to LLM or large multimodal scenarios. Code and checkpoints are not yet released, which limits immediate reproducibility.
>
> **A**: We conducted additional experiments on Llama-3.2-3B and Llama-3.1-8B models with RegMean and RegMean++. Fine-tuned checkpoints are employed from [10].
>
> Following [10], we merged five different candidates, each covers one of the following domains: instruction following, mathematics, multilingual understanding, coding, and safety. Accordingly, the merged models are evaluated on 11 benchmarks reflecting these domains, instruction following: IFEval [1], mathematics: GSM8K [2], multilingual understanding: Multilingual MMLU [3], Multilingual ARC [3], Multilingual Hellaswag [3], coding: Humaneval+ [4], MBPP+ [5], and safety: WildGuardTest [6], HarmBench [7], DoAnythingNow [8], XSTest [9].
>
> The two tables below show the results of merging candidates from Llama-3.2-3B and Llama-3.1-8B models. We perform grid search over different scaling factors $\alpha \in \{ 0.1, 0.3, 0.5, 0.7, 0.9 \}$, and report the best result based on the averaged multi-domain performance. For calculating the inner-product matrices, we use 256 samples for each domain with max sequence length of 2048, and batch size of 1. We find that RegMean++ outperforms RegMean when merging candidates of Llama-3.1-8B, while it lags behind RegMean when merging candidates of Llama-3.2-3B. However, RegMean++ slightly underperforms RegMean in the instruction following task. Overall, these results validate the versatility and effectiveness of RegMean++ as it is not confined to vision classification tasks; it can generally be applied to text generation tasks.
>
> *Code and Checkpoints*: We wish to clarify that reproducibility can be done immediately. We have already submitted the code in the supplemental material, as well as mentioned how to download the fine-tuned checkpoints in Appendix B.2. In the submitted code, we also described how to set up the environment for reproducing all of the reported results in this paper; the evaluation scripts are explicitly provided. As mentioned in Appendix B.2, we employ off-the-shelf fine-tuned checkpoints from the previous work [11]; by executing these evaluation scripts, all of these checkpoints will be automatically downloaded. We will release the merge checkpoints upon acceptance.
>
> *Llama-3.2-3B:*
>
> |Method| Instruction following |Math|Multilingual|Coding|Safety|Avg.|
> |-|-|-|-|-|-|-|
> |RegMean|8.3|35.5| 47.3| 39.2| 39.8| 34.0|
> |RegMean++|6.8|35.1| 47.4| 37.0| 38.5| 33.0|
>
> *Llama-3.1-8B:*
>
> |Method|Instruction following|Math|Multilingual|Coding|Safety|Avg.|
> |-|-|-|-|-|-|-|
> |RegMean|26.6|63.2|49.0|48.3|36.9|44.8|
> |RegMean++|11.1|65.8|53.1|52.3|46.3|45.7|
>
> [1] Zhou, Jeffrey, et al. "Instruction-following evaluation for large language models." arXiv preprint arXiv:2311.07911 (2023).
>
> [2] Cobbe, Karl, et al. "Training verifiers to solve math word problems." arXiv preprint arXiv:2110.14168 (2021).
>
> [3] Lai, Viet, et al. "Okapi: Instruction-tuned large language models in multiple languages with reinforcement learning from human feedback." EMNLP 2023.
>
> [4] Chen, Mark. "Evaluating large language models trained on code." arXiv preprint arXiv:2107.03374 (2021).
>
> [5] Austin, Jacob, et al. "Program synthesis with large language models." arXiv preprint arXiv:2108.07732 (2021).
>
> [6] Han, Seungju, et al. "Wildguard: Open one-stop moderation tools for safety risks, jailbreaks, and refusals of llms." NeurIPS 2024.
>
> [7] Mazeika, Mantas, et al. "HarmBench: A Standardized Evaluation Framework for Automated Red Teaming and Robust Refusal." ICML 2024.
>
> [8] Shen, Xinyue, et al. "" do anything now": Characterizing and evaluating in-the-wild jailbreak prompts on large language models." CCS 2024.
>
> [9] Röttger, Paul, et al. "Xstest: A test suite for identifying exaggerated safety behaviours in large language models." NAACL 2024.
>
> [10] He, Yifei, et al. "MergeBench: A Benchmark for Merging Domain-Specialized LLMs." arXiv preprint arXiv:2505.10833 (2025).
>
> [11] Tang, Anke, et al. "Fusionbench: A comprehensive benchmark of deep model fusion." arXiv preprint arXiv:2406.03280 (2024).
>
> ---
>
> We hope these address your concerns. If any questions remain, we are happy to discuss further!

---

### Official Review · Reviewer_cLJb · 2025-11-01

**Soundness:** 3
**Presentation:** 3
**Contribution:** 3
**Rating:** 6
**Confidence:** 5

**Summary:**

This paper presents RegMean++, an extension of RegMean that augments the merging objective with both intra-layer and cross-layer dependency modeling. The method is evaluated across diverse scenarios—ID and OOD settings, sequential merging, and merging with corrupted data—and is accompanied by a detailed layer-wise importance analysis of the merging process.

**Strengths:**

[1] The paper is clearly written. The idea of using representations from the merged model—rather than the originals—is a neat and effective trick that substantially boosts merging performance.

[2] The evaluation is comprehensive, covering multiple vision benchmarks and providing thoughtful analyses.

**Weaknesses:**

[1] The study is confined to vision tasks. Demonstrating results on LLMs—where model merging is widely practised—would significantly strengthen the paper’s impact.

[2] While RegMean++ advances over RegMean, the sequential-merging results should also be compared to state-of-the-art methods tailored for this setting to establish competitiveness.

**Questions:**

(1) The reported performance for SOTA baselines such as Iso-C [1] and TSV-M[2] appears considerably lower than in their original papers. Is this due to a different set of model checkpoints being used? If so, could you verify whether the same trends hold when evaluating on the exact checkpoints used in those works?

References:

[1] Marczak, D., Magistri, S., Cygert, S., Twardowski, B., Bagdanov, A. D., & van de Weijer, J. (2025). No task left behind: Isotropic model merging with common and task-specific subspaces. arXiv preprint arXiv:2502.04959.

[2] Gargiulo, A. A., Crisostomi, D., Bucarelli, M. S., Scardapane, S., Silvestri, F., & Rodola, E. (2025). Task singular vectors: Reducing task interference in model merging. In Proceedings of the Computer Vision and Pattern Recognition Conference (pp. 18695-18705).

**Details Of Ethics Concerns:**

No concerns.

---

> ### Author Response · Authors · 2025-11-22
> **Official Comments by Authors (1/2)**
>
> We thank the reviewer for the comments. Please see our responses to your concerns below.
>
> ---
>
> > **Q**: [1] The study is confined to vision tasks. Demonstrating results on LLMs—where model merging is widely practised—would significantly strengthen the paper’s impact.
>
> **A**: We thank the reviewer for this suggestion. We conducted additional experiments on Llama-3.2-3B and Llama-3.1-8B models with RegMean and RegMean++. Fine-tuned checkpoints are employed from [10].
>
> Following [10], we merged five different candidates, each covers one of the following domains: instruction following, mathematics, multilingual understanding, coding, and safety. Accordingly, the merged models are evaluated on 11 benchmarks reflecting these domains, instruction following: IFEval [1], mathematics: GSM8K [2], multilingual understanding: Multilingual MMLU [3], Multilingual ARC [3], Multilingual Hellaswag [3], coding: Humaneval+ [4], MBPP+ [5], and safety: WildGuardTest [6], HarmBench [7], DoAnythingNow [8], XSTest [9].
>
> The two tables below show the results of merging candidates from Llama-3.2-3B and Llama-3.1-8B models. We perform grid search over different scaling factors $\alpha \in \{ 0.1, 0.3, 0.5, 0.7, 0.9 \}$, and report the best result based on the averaged multi-domain performance. For calculating the inner-product matrices, we use 256 samples for each domain with max sequence length of 2048, and batch size of 1. We find that RegMean++ outperforms RegMean when merging candidates of Llama-3.1-8B, while it lags behind RegMean when merging candidates of Llama-3.2-3B. However, RegMean++ slightly underperforms RegMean in the instruction following task. Overall, these results validate the versatility and effectiveness of RegMean++ as it is not confined to vision classification tasks; it can generally be applied to text generation tasks.
>
> *Llama-3.2-3B:*
>
> |Method| Instruction following |Math|Multilingual|Coding|Safety|Avg.|
> |-|-|-|-|-|-|-|
> |RegMean|8.3|35.5| 47.3| 39.2| 39.8| 34.0|
> |RegMean++|6.8|35.1| 47.4| 37.0| 38.5| 33.0|
>
> *Llama-3.1-8B:*
>
> |Method|Instruction following|Math|Multilingual|Coding|Safety|Avg.|
> |-|-|-|-|-|-|-|
> |RegMean|26.6|63.2|49.0|48.3|36.9|44.8|
> |RegMean++|11.1|65.8|53.1|52.3|46.3|45.7|
>
> [1] Zhou, Jeffrey, et al. "Instruction-following evaluation for large language models." arXiv preprint arXiv:2311.07911 (2023).
>
> [2] Cobbe, Karl, et al. "Training verifiers to solve math word problems." arXiv preprint arXiv:2110.14168 (2021).
>
> [3] Lai, Viet, et al. "Okapi: Instruction-tuned large language models in multiple languages with reinforcement learning from human feedback." EMNLP 2023.
>
> [4] Chen, Mark. "Evaluating large language models trained on code." arXiv preprint arXiv:2107.03374 (2021).
>
> [5] Austin, Jacob, et al. "Program synthesis with large language models." arXiv preprint arXiv:2108.07732 (2021).
>
> [6] Han, Seungju, et al. "Wildguard: Open one-stop moderation tools for safety risks, jailbreaks, and refusals of llms." NeurIPS 2024.
>
> [7] Mazeika, Mantas, et al. "HarmBench: A Standardized Evaluation Framework for Automated Red Teaming and Robust Refusal." ICML 2024.
>
> [8] Shen, Xinyue, et al. "" do anything now": Characterizing and evaluating in-the-wild jailbreak prompts on large language models." CCS 2024.
>
> [9] Röttger, Paul, et al. "Xstest: A test suite for identifying exaggerated safety behaviours in large language models." NAACL 2024.
>
> [10] He, Yifei, et al. "MergeBench: A Benchmark for Merging Domain-Specialized LLMs." arXiv preprint arXiv:2505.10833 (2025).

---

> ### Author Response · Authors · 2025-11-22
> **Official Comments by Authors (2/2)**
>
> > **Q**: [2] While RegMean++ advances over RegMean, the sequential-merging results should also be compared to state-of-the-art methods tailored for this setting to establish competitiveness.
>
> **A**: We conducted sequential merging for all methods on the ViT-B-32 model. Following the setup described in our paper, we progressively merged four checkpoints at a time until all 20 tasks were merged for a predefined task sequence. The table below shows the performance of all methods, averaged over five different task sequences. We find that RegMean++ achieves superior performance after 8, 12, 16, and 20 tasks have been merged. Iso-C and Iso-CTS, although demonstrating strong performance on standard scenarios, these methods dramatically fail in sequential merging. Their performance shows a sharp decreasing trend after 12 tasks are merged.
>
> | Method                | 4 tasks merged | 8 tasks merged | 12 tasks merged | 16 tasks merged | 20 tasks merged |
> |-----------------------|----------------|----------------|-----------------|-----------------|-----------------|
> | Model Soups           | 58.9           | 59.3           | 60.1            | 59.1            | 60.0            |
> | Task Arithmetic       | 58.2           | 57.5           | 58.8            | 58.3            | 58.1            |
> | TIES-Merging          | 59.7           | 59.9           | 60.4            | 59.4            | 60.8            |
> | TSV-M                 | 56.9           | 61.4           | 62.0            | 64.0            | 63.5            |
> | DOGE TA               | 59.9           | 62.4           | 63.0            | 62.7            | 64.0            |
> | Iso-C                 | 59.2           | 60.7           | 58.7            | 53.4            | 44.4            |
> | Iso-CTS               | 58.7           | 57.8           | 47.9            | 32.8            | 28.3            |
> | Fisher Merging        | 58.9           | 59.5           | 59.7            | 58.9            | 60.4            |
> | RegMean               | **61.0**       | 64.7           | 65.1            | 65.7            | 66.9            |
> | **RegMean++** (Ours)  | 60.5           | **65.3**       | **65.7**        | **66.7**        | **68.7**        |
> | Layer-wise AdaMerging | 57.7           | 60.1           | 61.7            | 61.4            | 61.1            |
> | DOGE AM               | 57.0           | 61.2           | 62.4            | 63.9            | 63.1            |
>
> > **Q:** (1) The reported performance for SOTA baselines such as Iso-C [1] and TSV-M[2] appears considerably lower than in their original papers. Is this due to a different set of model checkpoints being used? If so, could you verify whether the same trends hold when evaluating on the exact checkpoints used in those works?
>
> **A**: This performance difference is due to the difference in checkpoints employed. We used the sets of checkpoints released by FusionBench [3], which are different from [1, 2].
>
> Following this suggestion, we reproduced TSV-M [2], Iso-C, Iso-CTS [1], RegMean, and our RegMean++ on the set of checkpoints provided by [1, 2] under the same setup described in the paper. The reproduced performance for the 8-task merging, as shown in the table below, matches the one reported in the respective papers.
>
> |Method|ViT-B/32 Reproduced|ViT-B/32 Reported [1, 2]|ViT-B/16 Reproduced|ViT-B/16 Reported [1, 2]|ViT-L/14 Reproduced|ViT-L/14 Reported [1, 2]|
> |-|-|-|-|-|-|-|
> |TSV-M| 85.66| 85.86| 88.97| 89.01| 92.94| 92.98|
> |Iso-C| 86.25| 86.30| 90.46| 90.60| 94.33| 94.20|
> |Iso-CTS| 86.37| 86.20| 91.16|91.10| 94.78| 94.70|
> |RegMean| 80.47| -| 83.35|-| 88.07|-|
> |RegMean++| 82.62| -| 85.41|-| 89.99| -|
>
> [1] Marczak, Daniel, et al. "No task left behind: Isotropic model merging with common and task-specific subspaces." ICML 2025.
>
> [2] Gargiulo, Antonio Andrea, et al. "Task singular vectors: Reducing task interference in model merging." CVPR 2025.
>
> [3] Tang, Anke, et al. "Fusionbench: A comprehensive benchmark of deep model fusion." arXiv preprint arXiv:2406.03280 (2024).
>
> ---
>
> We hope these address your concerns. If any questions remain, we are happy to discuss further!

---

### Author Response · Authors · 2025-12-02
**Summary of Our Responses during Rebuttal**

Dear Reviewers, ACs, SACs, and PCs,

We sincerely thank you for your time and effort in reviewing our paper.
We appreciate your constructive feedback and recognition of the strengths and contributions of our work.
Let us summarize as follows.

In this paper:

(1) We identify a critical limitation of the existing regression-mean (RegMean) model merging method: it merges each linear layer independently, overlooking how the features flow within the merge model and influence its final prediction.

(2) We introduce RegMean++ that explicitly incorporates both intra- and cross-layer dependencies of the merge model into RegMean’s objective. RegMean++ consistently improves over RegMean with stronger feature alignments.

(3) Through benchmarking against 11 advanced model merging methods, we show that RegMean++ achieves competitive or state-of-the-art performance in: in-domain merging, out-of-domain generalization, robustness, and sustainability to sequential merging or large-scale tasks.

(4) We conduct layer-wise analysis and observe: (i) for all existing merging methods, merging linear layers in only the middle and deep transformer layers preserves over 98\% of the performance compared to using all; (ii) merging linear layers in MLP modules consistently outperforms using those in the attention heads.


During rebuttal, our detailed responses have addressed individual concerns and suggestions from each Reviewer:
* **Additional experiments**: We conducted experiments on Llama-3.2-3B and Llama-3.1-8B for RegMean and RegMean++ across 11 benchmarks (Reviewer cLJb, sT1A, uBzW, and tzZH). We extended sequential-merging analysis to all baselines (Reviewer cLJb).
* **Result discrepancies**: Both Reviewer cLJb and Reviewer uBzW noticed an inconsistency with prior works. We clarified this with additional experiments on checkpoints provided by prior works.
* **Motivation, novelty, methodology, and contributions**: We re-clarified motivation, methodology's details, and contributions (Reviewer sT1A). We further highlighted the novelty of modeling the merge model's feature flow for regression-mean merging (Reviewer sT1A and uBzW).
* **Computational overhead**: We provided a detailed analysis of execution time and memory usage for RegMean++ and all baselines (Reviewer sT1A, uBzW, and tzZH).
* **Code and checkpoint release**: We confirmed code availability and merge checkpoint release plan (Reviewer sT1A).
* **OOD performance analysis**: We addressed the concern regarding out-of-distribution improvement (Reviewer uBzW).
* **Non-linearity in merging**: We clarified the handling of non-linearity in regression-mean merging and expanded discussion on this as our motivation (Reviewer tzZH).
* **Theoretical conditions**: We re-clarified the scenarios in which RegMean++ are expected to succeed or fail (Reviewer tzZH).

We regret that we were not able to engage in further discussion during the rebuttal phase.
We would be grateful if the Reviewers, ACs, and SACs could kindly consider our responses when making the final decision, as we believe we have adequately addressed all their concerns.
We still believe in a fair assessment.

We remain happy to address any further questions or suggestions after the final decision.

Once again, we thank you for your invaluable comments and suggestions, which have significantly improved our work.

Sincerely yours,

Authors of Paper 6631

---

### Meta-Review · Area_Chair_TfgW · 2026-01-01

**Summary:**

This paper proposes RegMean++, an extension of Regression Mean for model merging that recomputes merging statistics using merged-model activations in order to account for intra- and cross-layer dependencies. The authors argue that the original RegMean overlooks feature flow across layers and that this modification leads to improved performance. The method is evaluated extensively on vision benchmarks, including in-domain, out-of-domain, robustness, and sequential merging settings, and additional LLM experiments and efficiency analyses are provided during rebuttal.

The paper is generally well written and easy to follow, and the empirical evaluation is broad and carefully executed. Several reviewers acknowledge that the idea of using merged-model representations rather than candidate-model representations is intuitive and empirically effective. The experimental section is comprehensive, covering multiple architectures, task counts, and distribution shifts, and the authors are responsive during rebuttal, providing additional experiments on LLMs, sequential merging baselines, and computational cost analyses that improve the completeness of the submission.

Despite the solid empirical effort, the main concern is that the contribution remains incremental and lacks strong conceptual or theoretical grounding. RegMean++ largely modifies RegMean by changing how activations are collected, but does not introduce a new merging objective or principled framework for modeling cross-layer interactions. Claims about incorporating intra- and cross-layer dependencies are not supported by formal analysis, theoretical justification, or clear optimization objectives, and the rebuttal primarily relies on empirical improvements rather than addressing this gap. In addition, scalability and generality remain unclear: the added computation grows with depth, and the newly added LLM experiments show mixed results, with RegMean++ underperforming RegMean in some settings. Compared to recent model merging methods that introduce more substantial structural or geometric insights, the novelty of the approach appears limited.

Overall, while this is a careful and empirically strong extension of RegMean, the paper does not yet meet the bar for ICLR in terms of novelty and principled contribution. The gains over the baseline are incremental, the theoretical story remains underdeveloped, and the advantages over stronger state-of-the-art methods are not consistently demonstrated. Given the mixed reviewer sentiment and the remaining concerns after rebuttal, I lean toward reject.

**Reviewer Concerns:**

Some reviewer concerns were addressed in the rebuttal. The authors clarified discrepancies in baseline performance by reproducing results on original checkpoints, which resolves questions raised by multiple reviewers regarding fairness of comparison. They also added sequential merging results for additional baselines, provided runtime and memory analyses, and included experiments on LLMs, partially addressing concerns about evaluation scope and computational cost.

However, the main conceptual concerns remain unresolved. In particular, the issue of limited novelty and lack of theoretical grounding raised by Reviewer sT1A was not adequately addressed. The rebuttal reiterates empirical improvements but does not provide a principled objective or convincing explanation of how intra- and cross-layer dependencies are meaningfully modeled. Concerns about when and why the method improves over RegMean, especially for OOD generalization and large-scale settings, also remain largely qualitative. As a result, despite the additional experiments, core questions about the contribution and generality of the approach persist.

**Reviewer Scores:**

Reviewer cLJb: The rebuttal addressed concerns about baseline discrepancies and evaluation scope. However, given the reviewer’s original position, the score would likely remain at a similar marginally positive level.

Reviewer sT1A: The main concerns regarding limited novelty, lack of theoretical grounding, and unclear methodology were not resolved in the rebuttal. The reviewer’s strong reject score would likely remain unchanged.

Reviewer uBzW: While some empirical concerns were addressed, the core issue of limited novelty remains. This reviewer might slightly increase confidence in the results, but the score would likely stay marginally below the acceptance threshold.

Reviewer tzZH: Questions about computational cost and scalability were partially addressed, but conceptual concerns remain. The reviewer’s score would likely remain similar or decrease slightly.

---

### Decision · Program_Chairs · 2026-01-26

Reject